# NaviAgent: Bilevel Planning on Tool Navigation Graph for Large-Scale Orchestration

## Abstract

Large language models (LLMs) have recently demonstrated the ability to act as function call agents by invoking external tools, enabling them to solve tasks beyond their static knowledge. However, existing agents typically call tools step by step at a time without a global view of task structure. As tools depend on each other, this leads to error accumulation and limited scalability, particularly when scaling to thousands of tools. To address these limitations, we propose NaviAgent, a novel bilevel architecture that decouples task planning from tool execution through graph-based modeling of the tool ecosystem. At the task-planning level, the LLM-based agent decides whether to respond directly, clarify user intent, invoke a toolchain, or execute tool outputs, ensuring broad coverage of interaction scenarios independent of inter-tool complexity. At the execution level, a continuously evolving Tool World Navigation Model (TWNM) encodes structural and behavioral relations among tools, guiding the agent to generate scalable and robust invocation sequences. By incorporating feedback from real tool interactions, NaviAgent supports closed-loop optimization of planning and execution, moving beyond tool calling toward adaptive navigation of large-scale tool ecosystems. Experiments show that NaviAgent achieves the best task success rates across models and tasks, and integrating TWMN further boosts performance by up to 17 points on complex tasks, underscoring its key role in toolchain orchestration.

## 1 Introduction

Large language models (LLMs) are increasingly deployed as function call agents, moving beyond single utilities toward complex multi-stage workflows (Shen et al., 2023; Yang et al., 2023; Qu et al., 2025). However, real-world environments contain thousands of heterogeneous tools that are continually updated, while tasks demand long sequences of coordinated invocations. Agents built around fixed tool descriptions or rigid workflows fail to adapt, making API drift, continual updates, and unseen tool compositions key challenges for function call agents.

Existing approaches attempt to mitigate brittleness but remain incomplete. Some embed tool knowledge directly into model parameters (Wang et al., 2024), which reduces context demands but requires costly retraining when APIs change. Others derive static graphs from invocation logs (Liu et al., 2024b), yet sparse traces and missing parameter relations hinder generalization. Policy-adaptation methods adjust individual tools with feedback (Chen et al., 2024), while clustering-based planners enable substitutions (Liu et al., 2024c). Taken together, existing methods can be broadly categorized into two camps: either structured but static, failing to evolve with the ecosystem, or adaptive but unstructured, lacking the representations needed to capture composability and complementarity.

Underlying these challenges is the complexity of the tool ecosystem: it spans thousands of heterogeneous tools, exhibits interdependencies such as parameter flows and functional complementarity, and evolves continually through addition, update, and deprecation. Such properties reveal why step-by-step invocation without global awareness cannot achieve reliable tool composition. The difficulty is compounded by the fact that API documentation is written for humans and often misaligns with how models interpret and use individual tools (Qu et al., 2024), while flat catalogues provide little information on how tools compose, substitute, or adapt as the ecosystem changes. What is needed is a structured representation learned from execution traces that makes these dependencies explicit and continually adapts with feedback.

We propose **NaviAgent**, a bilevel planning framework that decouples high-level task reasoning from low-level execution. At the planning level, NaviAgent defines a four-dimensional decision space (direct response, intent clarification, toolchain retrieval, tool execution) covering core tool invocation scenarios, allowing the agent to operate without reasoning over complex inter-tool connections. At the execution level, it constructs the Tool World Navigation Model (TWNM), which encodes both structural and behavioral dependencies learned from execution traces. By coupling these graph-based representations with navigation strategies, TWNM enables retrieval, substitution, and multi-tool composition as the ecosystem evolves. Execution feedback continually updates both TWNM and the decision policy, forming a closed loop for robust adaptation to changing APIs.

Our main contributions are as follows: i) **NaviAgent Architecture.** The first bilevel agent framework that decouples high-level task planning from low-level tool execution, enabling scalable task composition across thousands of tools while preserving efficiency. ii) **Tool World Navigation Model.** A unified model that captures inter-tool structures and behavioral dependencies from execution traces, and supports navigation and flexible search in large-scale tool ecosystems. iii) **Closed-loop Evolution.** A feedback mechanism where execution traces continuously refine TWNM and decision strategies, driving the co-evolution of representation and decision-making.

## 2 RELATED WORK

**Single-Tool Invocation.** Early research focused on enhancing LLMs' single-tool invocation capabilities. TALM (Parisi et al., 2022) established foundational paradigms through predefined template chains, while EasyTool (Yuan et al., 2024) introduced structured tool descriptions to reduce semantic parsing overhead. For long-context scenarios, tool documentation compression techniques preserved critical semantics via summarization, enabling low-resource tool usage (Xu et al., 2024). Toolformer (Schick et al., 2023) innovatively embedded tool invocation APIs in pre-training, allowing self-supervised learning of usage patterns from unlabeled data. In multimodal settings, GPT4Tools (Yang et al., 2023) improved visual tool generalization (e.g. object detection) by aligning vision-language instructions with tool descriptions.

**Multi-Tool Orchestration.** As tool libraries expanded, HuggingGPT (Shen et al., 2023) proposed a four-stage pipeline (plan, select, execute, respond) for standardized multi-tool workflows, while Chameleon (Lu et al., 2023) integrated heterogeneous tools (13+ types) via modular composition. Similarly, $\alpha$-UMI (Shen et al., 2024) decomposes the tool-use process into planning, invocation, and summarization, but uniquely assigns each stage to a dedicated lightweight LLM, enabling modular updates and improved performance, especially for smaller models. For small toolkits, TRICE (Qiao et al., 2023) optimized single tool policies via execution feedback, and ToolFactory (Ni et al., 2025) automated tool adaptation through domain-guided code synthesis. However, these approaches struggled with dynamic collaboration. For large-scale toolkits, Confucius (Gao et al., 2024) addressed combinatorial explosion via hierarchical tool classification, and ToolVerifier (Mekala et al., 2024) improved selection robustness through self-verification mechanisms.

**Dynamic Planning & Adaptation.** Static frameworks faltered under open-domain task complexity, prompting dynamic decision mechanisms. ReAct (Yao et al., 2023b) pioneered the decoupling of reasoning from tool calls through chain-of-thought planning. Building on this, Reflexion (Shinn et al., 2023) enhanced error recovery by introducing iterative self-reflection, significantly improving fault tolerance in complex tasks. For long-horizon tasks, path search techniques became pivotal: Tree-of-Thoughts (ToT) (Yao et al., 2023a) formalized tool invocation as searchable reasoning trees with dynamic branching, while ToolLLM (Qin et al., 2023) optimized search efficiency through functional hierarchy-guided DFS. ToolChain (Zhuang et al., 2023) further advanced this by employing heuristic cost estimation to prioritize high-success-rate branches. Yet, these methods assumed static tool relationships, failing to adapt to API drift or cross-domain tasks. ControlLLM (Liu et al., 2024d) built static dependency graphs for task decomposition, whereas ToolNet (Liu et al., 2024b) dynamically updated tool relations from historical calls, both limited by sparse multi-hop interaction data. This gap motivates our TWNM that jointly models structural dependencies and behavioral adaptations to capture evolving tool relations, aligning with findings that graph learning enhance LLM planning (Wu et al., 2024; Besta et al., 2024).

# 3 METHODOLOGY

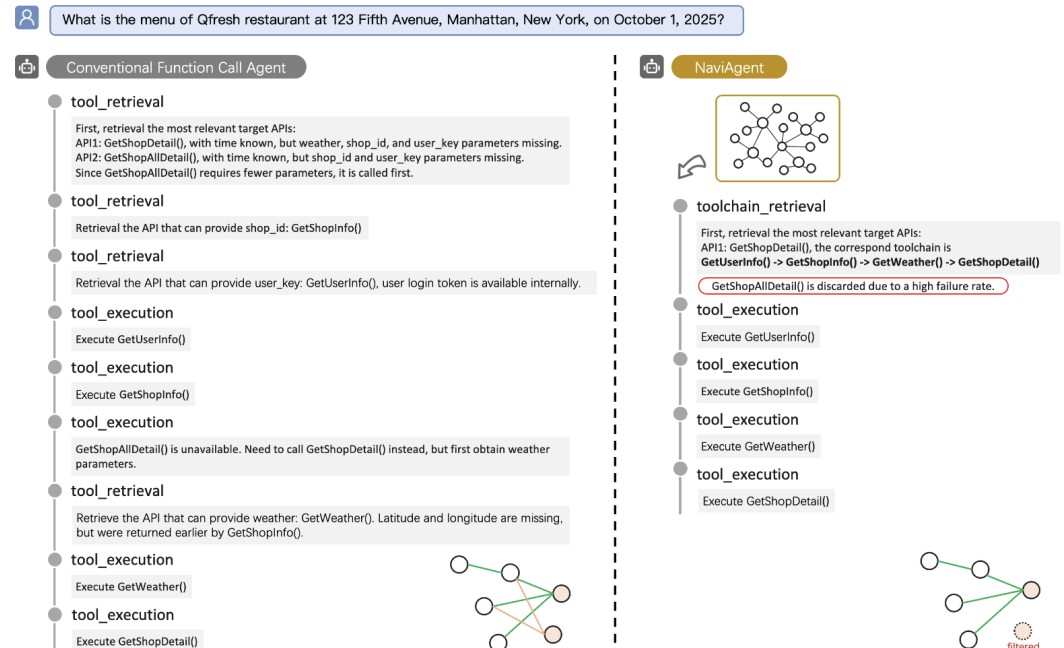

Figure 1: Conventional function call agents vs. NaviAgent.

A key challenge in function call agents is that API calls usually have interdependent parameters and strict invocation orders. NaviAgent addresses this challenge by retrieving the entire toolchain before execution, rather than calling APIs step by step. This global reasoning allows the agent to plan an end-to-end path from the user query to the target API once and execute it directly. As illustrated in Figure 1, NaviAgent avoids repeated retrievals and automatically discards unreliable APIs, leading to more efficient and robust execution.

## 3.1 A FOUR-DIMENSIONAL DECISION AGENT

### 3.1.1 DEFINITION

The architecture achieves end-to-end decision-making through LLMs, formally modeled as a quintuple $(\mathcal{H}, \mathcal{O}, \mathcal{G}, \mathcal{A}, F)$ where $\mathcal{H} = \{(o_{t-i}, a_{t-i})\}_{i=1}^{n}$ represents historical states (containing state sequence $\{o_i\}$ and action sequence $\{a_i\}$), $\mathcal{O}$ denotes the observation, $\mathcal{G}$ represents the tool dependency graph, $\mathcal{A} = \{Direct\ Response,\ Intent\ Clarification,\ ToolChain\ Retrieval,\ Tool\ Execution\}$ defines the four dimensional decision space, where each action corresponds to directly answering the user, requesting clarification, retrieving candidate tool dependency subgraph via graph pruning, or execute selected toolchains, respectively. $F : \mathcal{H} \times \mathcal{O} \times \mathcal{G} \to \mathcal{A}$ specifies the decision function. At each time step $t$, the agent constructs its decision context as follows. The historical context $\mathcal{H}_t$ is defined as

$$\mathcal{H}_t = \langle (o_{t-3}, a_{t-3}), \dots, (o_{t-1}, a_{t-1}) \rangle \tag{1}$$

where a sliding window maintains the most recent three[1] observation-action pairs, capturing the agent's recent decision trajectory. The pruned tool dependency subgraph $\mathcal{G}'_{t-1} = (V, E, W)$ is computed from the agent's state at the previous time step $t-1$, where $V$ is the node set, $E$ is the edge set, and $W$ denotes the edge weights indicating dependency strengths. The subgraph is serialized into a tree-structured textual format, ensuring a simplified yet sufficient representation for selected toolchains. The overall decision function is then formulated as

$$a_t = F(\mathcal{H}_t, \mathcal{O}_t, \mathcal{G}'_{t-1}) \tag{2}$$

---

[1]Our experiments demonstrate that utilizing the most recent three observation-action pairs achieves the best balance between accuracy and efficiency.

where $\mathcal{O}_t$ is the current observation, and $a_t \in \mathcal{A}$ is the action selected at time $t$.

### 3.1.2 MODEL TRAINING

For supervised fine-tuning, we adopt the standard language modeling objective, computing the loss exclusively over the response or action generation segments. During training, the LLM-based agent receives as input the most recent historical state-action pairs $\mathcal{H}_t$, the current observation $\mathcal{O}_t$, and the pruned tool dependency subgraph $\mathcal{G}_{\mathrm{sub}}$. The model is trained to maximize the likelihood of the ground-truth action $a_t^*$ at step $t$, which is derived from high-quality, curated datasets (see Appendix E.2 for details):

$$\mathcal{L}_{\mathrm{SFT}} = -\frac{1}{N} \sum_{i=1}^{N} \log p_\theta(a_t^* \mid \mathcal{H}_t, \mathcal{O}_t, \mathcal{G}_{\mathrm{sub}}) \tag{3}$$

where $N$ is the number of training samples and $p_\theta$ denotes the agent's predicted probability over the action space.

### 3.2 TOOL WORLD NAVIGATION MODEL

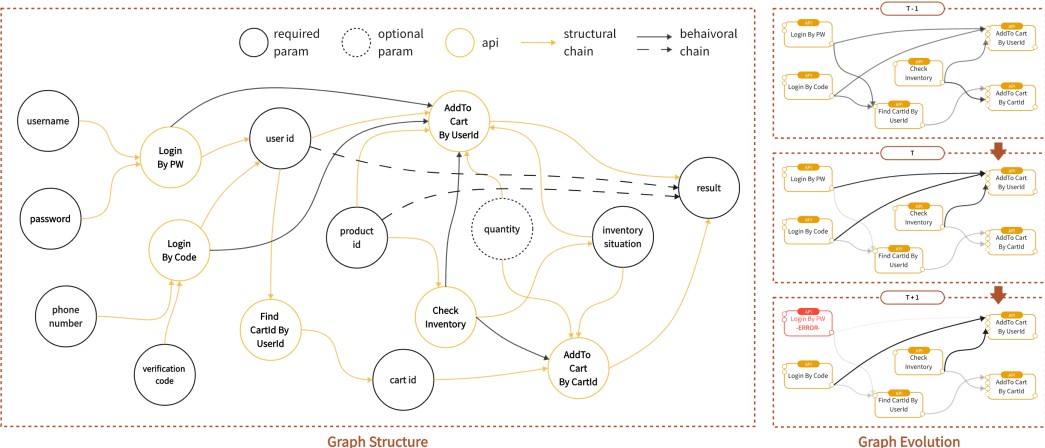

Figure 2: Tool dependency graph and its temporal evolution in TWNM. The left part shows the overall dependency relations, while the right part illustrates the pruning and evolution of executable subgraphs across time steps.

### 3.2.1 GRAPH CONSTRUCTION AND REPRESENTATION

While tool standardization frameworks (e.g., Anthropic's MCP) help normalize basic API metadata, challenges remain due to inconsistent parameter naming and undocumented tool dependencies. In our framework, each tool consists of one or more APIs. We address these issues by applying semantic similarity clustering to unify functionally equivalent parameters (see details in Appendix A).

**Definition.** We construct a directed weighted graph $\mathcal{G} = (V, E, W)$ with API and parameter nodes. Edges include structural chains, defined by API schemas (e.g., parameter-to-API and API-to-parameter connections), as well as behavioral chains, derived from historical usage data (e.g., API-to-API and parameter-to-parameter dependencies), as illustrated in Figure 2 (left). Each edge is assigned a statistical weight $\tilde{w}_{ij}$ reflecting empirical invocation patterns.

$$\tilde{w}_{ij} = \frac{N(v_i \rightarrow v_j)}{N(v_j)} \tag{4}$$

where $N(v_i \rightarrow v_j)$ counts the number of successful invocations from $v_i$ to $v_j$, and $N(v_j)$ is the total number of invocations involving $v_j$.

We formulate tool dependency discovery as a link prediction problem Hamilton et al. (2017); Zhang & Chen (2018); Zhou et al. (2020); Wu et al. (2024). To model this, we employ a Heterogeneous

Graph Transformer (HGT) that integrates node-level feature fusion, type-specific encoding, and relation-aware message passing. Each node is initialized with both semantic (BGE-based) and structural features (including invocation statistics and degree information), and projected into a unified embedding space. We stack two multi-head HGT layers to aggregate information from the 2-hop neighborhood. Notably, the attention mechanism incorporates a statistical edge weight $\tilde{w}_{uv}$ to reflect empirical call patterns:

$$\alpha_{uv}^{(k,r)} = softmax_{u \in \mathcal{N}_r(v)} \left( \frac{(\mathbf{W}_Q^{(k,r)} \mathbf{h}_u')^\top (\mathbf{W}_K^{(k,r)} \mathbf{h}_v')}{\sqrt{d_k}} + \mathbf{b}_r^{(k)} + \tilde{w}_{uv} \right) \tag{5}$$

where $\mathcal{N}_r(v)$ denotes the set of neighbors of node $v$ under relation $r$, and $\mathbf{h}_u'$, $\mathbf{h}_v'$ are the type-specific encoded representations of nodes $u$ and $v$ (see Appendix B.1 for details). $\mathbf{W}_Q^{(k,r)}$ and $\mathbf{W}_K^{(k,r)}$ are the query and key projection matrices for head $k$ and relation $r$, $\mathbf{b}_r^{(k)}$ is an edge-type-specific bias, and $d_k = d/8$ is the dimension per head. Then the concatenated head outputs are projected to obtain the final node embeddings, which are then used for link prediction.

### 3.2.2 GRAPH TRAINING OBJECTIVE

The graph model is trained with a hybrid loss that combines cross-entropy and adaptive margin objectives, both leveraging edge weights $\tilde{w}_{uv}$ to capture graded dependencies.

**Cross-entropy.**

$$\mathcal{L}_{\text{CE}} = -\frac{1}{|\mathcal{E}|} \sum_{(u,v) \in \mathcal{E}} \left[ \tilde{w}_{uv} \log p_{uv} + (1 - \tilde{w}_{uv}) \log(1 - p_{uv}) \right] \tag{6}$$

where $p_{uv}$ is the predicted link probability, $\tilde{w}_{uv}$ is the statistical edge weight serving as a soft label, and $\mathcal{E}$ denotes the set of all edges in the graph.

**Adaptive margin.** It assigns larger separation to higher-weight edges(i.e., $\tilde{w}_{uv} \to 1$), focusing learning on critical dependencies. For each positive edge $(u,v)^+ \in \mathcal{E}^+$, $k$ negative edges $\{(u_j, v)\}_{j=1}^k$ are sampled to construct positive and negative pairs for the margin loss.

$$m_{uv} = m_0 \left( 1 + \sigma(\tilde{w}_{uv}) \right) \tag{7}$$

$$\mathcal{L}_{\text{Margin}} = \frac{1}{|\mathcal{E}^+|} \sum_{(u,v)^+ \in \mathcal{E}^+} \frac{1}{k} \sum_{j=1}^k \left[ m_{uv} - s(u,v)^+ + s(u_j, v)^- \right]_+ \tag{8}$$

where $m_0$ is a base margin, $\sigma(\cdot)$ denotes the sigmoid function, $\tilde{w}_{uv}$ is the statistical edge weight, $s(u,v)$ measures the embedding similarity, $(u,v)^+$ represents a positive edge, $(u_j, v)^-$ denotes a negative sample, and $[\cdot]_+$ is the hinge function, and $\mathcal{E}^+$ is the set of positive edges.

The final training objective is a weighted sum of the two losses:

$$\mu_t = \mu_0 \cdot \gamma^t, \ \gamma \in (0, 1) \tag{9}$$

$$\mathcal{L} = \mu_t \cdot \mathcal{L}_{CE} + (1 - \mu_t) \cdot \mathcal{L}_{\text{Margin}} \tag{10}$$

where $\mu_t$ is the weight for the cross-entropy loss at epoch $t$, $\mu_0$ is the initial weight, and $\gamma$ is a decay factor controlling the rate at which the contribution of the cross-entropy loss decreases over training. This curriculum strategy first emphasizes accuracy, then discrimination, yielding accurate predictions and structured embeddings.

### 3.2.3 GRAPH SEARCH

At inference time, the predicted link probabilities $p_{uv}$ are used as edge weights $w_{uv}$ in the tool graph, forming the basis for weighted-graph search and toolchain planning. We adopt two representative search strategies adapted to this setting: an Alpha-Beta Pruning method that eliminates weak toolchains using dynamic thresholds, and a heuristic search that evaluates candidate toolchains with a composite fitness balancing connectivity, depth, and cumulative weights. Complete algorithms and parameter details are provided in Appendix B.2.

### 3.2.4 GRAPH EVOLUTION

The tool world is inherently dynamic, evolving as new tools are introduced, obsolete ones are deprecated, and usage patterns shift. As shown at the right of Figure 2: At time $T - 1$, selectable paths are relatively uniform (similar line shades), indicating multiple equally viable routes. By $T$, the TWNM has learned to prefer a more optimal path (the top, darker line). At $T + 1$, when the upper-left API becomes unavailable, the TWNM adapts by recomputing and selecting the lower route as the new optimal path. This demonstrates the TWNM's ability to learn from feedback and flexibly adjust its planning in response to runtime changes. To systematically support such adaptability, we design a graph evolution framework with three key mechanisms:

**Incremental Node Integration.** To accommodate newly introduced tools, we incrementally add new nodes via semantic similarity clustering, initializing their parameters (e.g., $N_{succ}(v) = 0$, $N_{fail}(v) = 0$ for successful and failed invocation counts) and the statistical weights of associated edges (e.g., $\tilde{w}_{uv} = 0$) to ensure consistency with existing graph features.

**Targeted Subgraph Pruning.** Obsolete or rarely used tools are selectively pruned based on a weighted combination of failure rates and invocation frequencies:

$$\text{Prune}(v) \propto \lambda \cdot \sigma(f_{fail}(v)) + (1 - \lambda) \cdot \sigma(f_{freq}(v)^{-1}) \tag{11}$$

where $\lambda \in [0, 1]$ controls the trade-off between failure rates and invocation frequencies, and $f_{fail}$ and $f_{freq}$ denote failure rates and invocation frequencies, respectively.

**Edge Attribute Propagation.** Long-term stability and short-term adaptation are balanced by updating the statistical edge weights $\tilde{w}_{uv}$ through a combination of historical trends and recent invocation success rates:

$$\tilde{w}_{uv}^{(t)} = \eta \cdot \underbrace{\tilde{w}_{uv}^{(t-1)}}_{\text{long}-\text{term weight}} + (1 - \eta) \cdot \underbrace{\frac{N_{succ}^{\text{recent } \tau \text{ days}}(u \to v)}{N_{succ}^{\text{recent } \tau \text{ days}}(v)}}_{\text{recent success rate}} \tag{12}$$

where $\eta \in [0, 1]$ balances long-term memory and recent observations, and $N_{succ}^{\text{recent } \tau \text{ days}}$ denotes successful invocations within a sliding window of $\tau$ days. These dynamically updated statistical edge weights $\tilde{w}_{uv}$ are subsequently used as soft labels for supervising model training, as described in Section 3.2.2.

## 3.3 DYNAMIC EXECUTION & PATH RECOMBINATION

Robust and adaptive toolchain orchestration is achieved through a bilevel dynamic planning framework, in which the agent manages action selection and the TWNM is responsible for toolchain planning.

**NaviAgent Workflow.** When a user query arrives, NaviAgent decides whether it can respond directly, clarify the user's intent, or rely on external tools. For more complex queries, NaviAgent decomposes the task into sub-tasks and categorizes them into two types: those that can be answered or clarified immediately, and those that demand toolchain retrieval. Unlike traditional agents that fetch tools sequentially, NaviAgent searches the existing tool dependency graph for a task-relevant subgraph and selects a feasible execution path for subsequent execution. More detailed cases can be found in the Appendix C.

**Path Recombination.** During execution, if a *tool execution* action fails due to an an API is unavailable or malfunctioning, the agent switches from execution to *toolchain retrieval* and invokes its TWNM module. As shown in Figure 3, TWNM searches the current tool dependency graph to recombine nodes and identify an alternative toolchain, which the agent then executes. This adaptive loop can be repeated until completion or infeasibility, enabling dynamic path recombination [2] that improves robustness and task success in complex tool environments.

---

[2]The recombination process terminates after at most four iterations (i.e., four recombination steps).

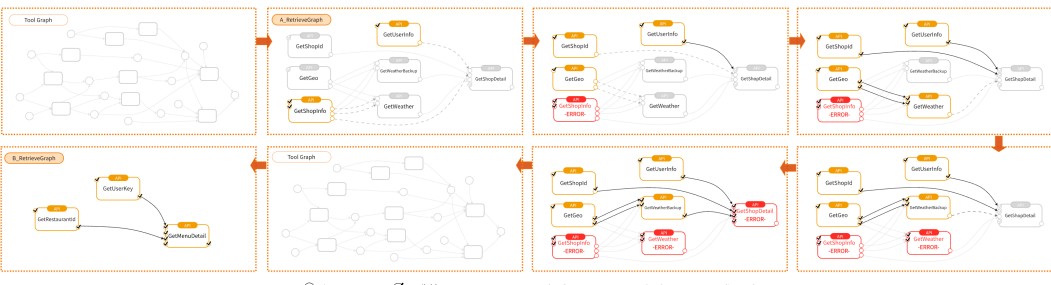

Figure 3: Representative Modes and Example of Path Recombination. i) **Direct substitution**: replacing a failed API with another offering the same functionality and compatible I/O schema (e.g., GetWeather→GetWeatherBackup); ii) **Common-output substitution**: combining multiple APIs whose outputs jointly meet the same intermediate requirement (e.g., GetShopInfo→GetShopId,GetGeo); iii) **Goal-level replanning**: retrieving a new graph to reach the final goal through an alternative route when the target API is infeasible (e.g., GetShopDetail→GetMenuDetail).

## 4 EXPERIMENTS

### 4.1 EXPERIMENTAL SETTINGS

**Datasets.** Our experiments are based on two public API benchmarks: API-Bank Li et al. (2023) and ToolBench Qin et al. (2023). As real-time API execution is currently unavailable, evaluation tasks are constructed in a simulated environment based on the extensive API lists and conversational trajectories provided by these datasets. Tasks are categorized into three levels of complexity: **Easy** (at most one API call or directly answerable), **Medium** (two API calls), and **Hard** (three or more APIs). Details of task generation are provided in Appendix D. For model fine-tuning, Qwen2.5-14B is trained on 3,500+ examples sampled from our generated task set, with strict separation between fine-tuning and evaluation data to prevent leakage.

**Baselines and Models.** The evaluation considers frameworks for real-world tool invocation, where managing large tool sets and enabling autonomous planning are critical. We select representative baselines in three major categories: (i) ReAct-based single-agent frameworks, where ReAct Yao et al. (2023b) serves as the foundational approach alternating reasoning and tool use; (ii) enhanced single-agent frameworks, where ToolLLM Qin et al. (2023) incorporates DFSDT-based planning with a dynamic backtracking mechanism; and (iii) multi-agent frameworks, where $\alpha$-UMI Shen et al. (2024) organizes modular tool-use stages via lightweight LLMs. Experiments are conducted across multiple foundation models, including open-source models (Qwen2.5-14B Yang et al. (2024), Qwen2.5-32B Tahmid & Sarker (2024), DeepSeek-R1-Distill-Qwen-32B(DeepSeek-R1-32B) Guo et al. (2025)) and closed-source models (DeepSeek-V3 Liu et al. (2024a), GPT-4o Hurst et al. (2024)), as well as a fine-tuned lightweight model (Qwen2.5-14B).

**Metrics.** Our evaluation framework considers three metrics: task success rate (TSR), execution steps (Steps), and task completion rate (TCR). TSR and Steps are the primary indicators, with TSR measuring output quality by evaluating whether the system's response fully satisfies the user's request (via LLM-based comparison with the ground truth), and Steps reflecting execution efficiency as the total number of LLM calls required to solve a task, counted only for successfully completed tasks. TCR serves as a supplementary metric, indicating whether the system produces a final output without prematurely terminating. Tasks are considered incomplete if they exceed the maximum allowed attempts, encounter parsing errors, or fail due to input token limits. Both TCR and TSR are reported as percentages over all evaluation tasks. All experiments details of training and inference setup provided in Appendix E.2.

## 4.2 RESULTS

In this section, we present the main results on ToolBench, comparing NaviAgent with strong baselines across various model sizes and task difficulties.

| Model | Method | Easy | | | Medium | | | Hard | | | All | | |
|---|---|---|---|---|---|---|---|---|---|---|---|---|---|
| | | TCR | TSR | Steps | TCR | TSR | Steps | TCR | TSR | Steps | TCR | TSR | Steps |
| Qwen2.5-14B | ReAct | 32.4 | 26.3 | 3.52 | 24.5 | 16.8 | 3.67 | 24.8 | 20.0 | 3.64 | 27.1 | 20.6 | 3.61 |
| | ToolLLM | 56.1 | 30.4 | 4.01 | 53.8 | 19.7 | 4.02 | 38.1 | 11.4 | 4.06 | 51.0 | 21.3 | 4.03 |
| | α-UMI | 77.7 | 39.2 | 5.53 | 77.9 | 25.0 | 5.88 | 67.6 | 13.3 | 6.07 | 75.5 | 26.9 | 5.74 |
| | **Dynamic+H** | 64.2 | **50.3** | 4.18 | 60.1 | **32.3** | 4.38 | 61.1 | **22.4** | 4.68 | 61.6 | **35.8** | 4.38 |
| Qwen2.5-32B | ReAct | 33.1 | 25.0 | 3.50 | 35.6 | 24.5 | 3.60 | 30.5 | 19.0 | 3.95 | 33.6 | 23.4 | 3.63 |
| | ToolLLM | 40.5 | 31.8 | 3.67 | 48.6 | 30.3 | 3.85 | 49.5 | 23.8 | 4.10 | 46.2 | 29.3 | 3.83 |
| | α-UMI | 78.4 | 49.3 | 5.66 | 78.8 | 26.0 | 6.02 | 77.1 | 22.9 | 6.58 | 78.3 | 32.8 | 5.94 |
| | **Dynamic+H** | 88.1 | **61.1** | 4.29 | 81.7 | **41.5** | 4.60 | 79.4 | **30.8** | 5.31 | 83.2 | **45.4** | 4.66 |
| Deepseek-V3 | ReAct | 46.6 | 36.5 | 3.52 | 58.7 | 38.5 | 3.50 | 48.6 | 23.8 | 3.74 | 52.5 | 34.5 | 3.54 |
| | ToolLLM | 56.2 | 47.4 | 3.80 | 58.8 | 30.0 | 3.92 | 29.7 | 24.8 | 3.90 | 51.3 | 34.4 | 3.86 |
| | α-UMI | 80.8 | 59.7 | 5.95 | 89.4 | 32.9 | 5.95 | 73.0 | 29.5 | 6.64 | 82.9 | 40.7 | 6.06 |
| | **Dynamic+H** | 97.9 | **71.8** | 4.40 | 96.3 | **48.5** | 4.45 | 97.0 | **44.9** | 5.19 | 97.0 | **55.2** | 4.60 |

Table 1: **Comparison of Baseline Frameworks on ToolBench.** TCR and TSR are reported as percentages (%), and lower Steps indicates higher efficiency. The best results are marked in **bold** and the second-best results are marked with underline.

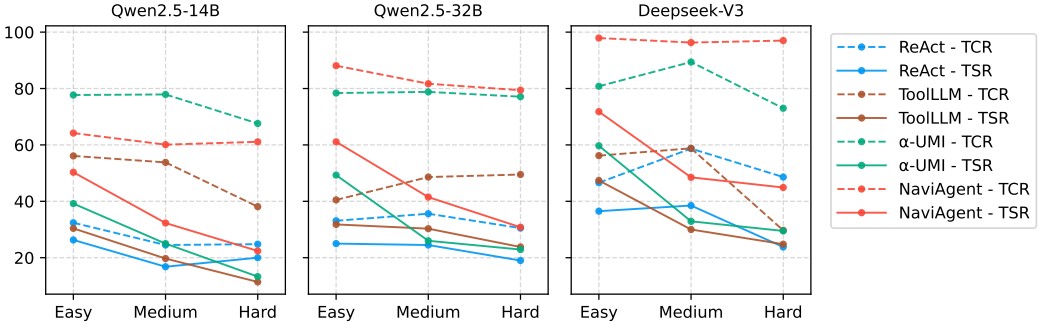

Figure 4: Evaluation of Frameworks on ToolBench Across Task Complexity.

**Overall Performance and Efficiency.** As shown in Table 1 and Figure 4, NaviAgent consistently achieves the highest TSR across all foundation models and task complexities, with absolute values of 35.8% on Qwen2.5-14B, 45.4% on Qwen2.5-32B, and 55.2% on Deepseek-V3. Compared to the average performance of the baselines on all tasks, NaviAgent achieves substantial gains of 12.9, 16.9, and 18.7 percentage points on Qwen2.5-14B, Qwen2.5-32B, and Deepseek-V3, respectively. Meanwhile, its execution steps remain close to those of the most efficient baseline, with differences typically within one step, thereby maintaining a strong balance between solution quality and execution efficiency. Consistent performance is also observed in our real-world API tests, with detailed results provided in Appendix F.1.

**Relative Improvement and Robustness.** NaviAgent achieves an average TSR improvement of over 10 percentage points compared to α-UMI, the strongest among the three baselines, across all difficulty levels, with the most significant gain of 15.4 percentage points on Deepseek-V3 for Hard tasks. We further observe that the relative drop in TSR from Easy to Hard tasks is substantially smaller for NaviAgent than for most baselines, particularly on larger foundation models. For example, on Deepseek-V3, NaviAgent's TSR decreases by only 37.5% from Easy to Hard, while ToolLLM and α-UMI experience drops of 47.7% and 50.6%, respectively.

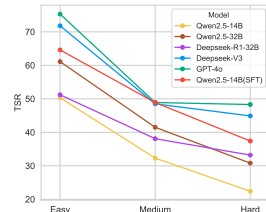

Figure 5: Effect of SFT on TSR.

| Model | Method | Easy | | | Medium | | | Hard | | | All | | |
|---|---|---|---|---|---|---|---|---|---|---|---|---|---|
| | | TCR | TSR | Steps | TCR | TSR | Steps | TCR | TSR | Steps | TCR | TSR | Steps |
| Qwen2.5-14B | Base | 46.4 | 36.0 | 5.38 | 50.5 | 22.9 | 5.39 | 62.0 | 16.3 | 5.76 | 51.8 | 25.6 | 5.47 |
| | Static+A | 57.3 | 43.7 | 4.37 | 61.2 | 29.0 | 4.54 | 53.0 | 14.0 | 4.59 | 58.1 | 30.3 | 4.50 |
| | Dynamic+A | 58.8 | 48.0 | 4.31 | 61.5 | 31.7 | 4.49 | 53.3 | 16.2 | 4.61 | 58.8 | 33.4 | 4.46 |
| | **Dynamic+H** | 64.2 | **50.3** | 4.18 | 60.1 | **32.3** | 4.38 | 61.1 | **22.4** | 4.68 | 61.6 | **35.8** | 4.38 |
| Qwen2.5-32B | Base | 77.7 | 47.7 | 5.42 | 75.8 | 32.7 | 6.00 | 86.9 | 19.0 | 7.04 | 78.9 | 34.4 | 6.05 |
| | Static+A | 82.8 | 50.7 | 4.47 | 83.3 | 40.6 | 5.07 | 79.7 | 26.3 | 5.30 | 82.3 | 40.6 | 4.93 |
| | Dynamic+A | 83.1 | 51.4 | 4.41 | 85.1 | 41.3 | 5.03 | 80.0 | **31.4** | 5.37 | 83.3 | 42.3 | 4.91 |
| | **Dynamic+H** | 88.1 | **61.1** | 4.29 | 81.7 | **41.5** | 4.60 | 79.4 | 30.8 | 5.31 | 83.2 | **45.4** | 4.66 |
| Deepseek-R1-32B | Base | 89.5 | 32.2 | 6.16 | 85.0 | 25.8 | 6.64 | 88.6 | 19.7 | 6.65 | 87.3 | 26.5 | 6.49 |
| | Static+A | 92.3 | 45.8 | 5.14 | 92.5 | 35.8 | 5.39 | 91.5 | 20.8 | 5.99 | 92.2 | 35.6 | 5.45 |
| | Dynamic+A | 92.6 | 51.4 | 5.06 | 93.3 | 38.0 | 5.33 | 91.4 | 21.9 | 5.93 | 92.6 | 38.6 | 5.38 |
| | **Dynamic+H** | 93.5 | 51.2 | 4.82 | 92.4 | 38.1 | 5.23 | 87.8 | **33.2** | 5.46 | 91.7 | **41.2** | 5.15 |
| Deepseek-V3 | Base | 93.7 | 66.3 | 5.26 | 93.8 | 39.7 | 6.00 | 94.7 | 31.1 | 6.22 | 94.0 | 46.3 | 5.81 |
| | Static+A | 92.9 | 70.5 | 4.31 | 95.8 | 47.4 | 4.66 | 93.4 | 31.1 | 5.05 | 94.3 | 51.1 | 4.64 |
| | Dynamic+A | 93.2 | 71.6 | 4.36 | 95.7 | 50.5 | 4.68 | 93.3 | 33.3 | 4.97 | 94.4 | 53.4 | 4.64 |
| | **Dynamic+H** | 97.9 | **71.8** | 4.40 | 96.3 | 48.5 | 4.45 | 97.0 | **44.9** | 5.19 | 97.0 | **55.2** | 4.60 |
| GPT-4o | Base | 92.0 | 62.7 | 5.07 | 91.0 | 35.2 | 5.67 | 94.5 | 27.8 | 6.26 | 92.1 | 42.3 | 5.61 |
| | Static+A | 99.5 | 72.1 | 4.21 | 98.3 | 43.6 | 5.35 | 97.8 | 37.9 | 5.85 | 98.6 | 51.5 | 5.10 |
| | Dynamic+A | 99.9 | **76.4** | 4.18 | 99.5 | 45.3 | 5.40 | 98.1 | 41.4 | 5.92 | 99.3 | 54.4 | 5.13 |
| | **Dynamic+H** | 99.6 | 75.3 | 4.01 | 94.5 | **48.9** | 4.71 | 98.9 | **48.3** | 5.12 | 97.1 | **57.2** | 4.58 |
| Qwen2.5-14B(SFT) | Base | 70.9 | 49.1 | 5.94 | 72.8 | 42.1 | 5.94 | 71.0 | 24.5 | 6.99 | 71.8 | 40.3 | 6.18 |
| | Static+A | 84.6 | 61.4 | 4.50 | 78.1 | 38.6 | 4.69 | 77.8 | 35.6 | 5.65 | 80.1 | 45.3 | 4.85 |
| | Dynamic+A | 85.8 | **64.9** | 4.58 | 78.4 | 39.9 | 4.75 | 78.1 | **39.0** | 5.59 | 80.7 | 47.7 | 4.89 |
| | **Dynamic+H** | 82.7 | 64.6 | 4.59 | 81.4 | **48.9** | 4.67 | 78.5 | 37.4 | 5.74 | 81.2 | **51.3** | 4.89 |

Table 2: **Impact of Naviagent Variants on ToolBench.** Base retains only the core agent; StaticA augments with a static graph without historical invocation data and Alpha-Beta pruning; DynamicA augments with a dynamic graph and Alpha-Beta pruning; DynamicH augments with a dynamic graph and heuristic pruning, which corresponds to our proposed NaviAgent. Metrics are reported as in Table 1. We also evaluate runtime, with detailed results reported in Appendix F.2.

**Adaptability through Fine-tuning.** Notably, with supervised fine-tuning, the smaller Qwen2.5-14B model achieves performance comparable to the larger 32B model (TCR 81.2% vs 83.2%, TSR 51.3% vs 45.4%, see Figure 5 and Table 2, D+N(Heur) row), indicating that fine-tuning can effectively close the gap between model sizes.

## 4.3 ABLATION STUDY

To further validate the effectiveness of each component in our framework, we conduct two sets of ablation studies.

**Effect of base Components.** We analyze the NaviAgent (Base) configuration, focusing on its four-dimensional decision space in successful ToolBench cases with the Deepseek-V3 model. Specifically, we categorize the proportion of cases resolved via **Clarification** (*intent clarification* to seek additional details from the user), **Re-retrieval** (recovering from initial *toolchain retrieval* failures by invoking alternative APIs), and **Normal** (tasks completed successfully in a single attempt without clarification or re-retrieval). Results are summarized in Figure 6, demonstrating that the four-dimensional decision space of the agent enables robust error recovery and flexible intent handling, contributing to overall performance gains.

**Effect of TWNM Components.** Table 2 shows clear gains from each design choice. Compared with the Base (agent only), NaviAgent (DynamicH) improves TSR by +11.8 points on average, confirming the value of graph-based planning with search. Dynamic graphs further outperform static ones on hard tasks (e.g., +5.1 on Qwen2.5-32B, +2.0 on GPT-4o), and replacing Alpha-Beta with heuristic search brings the best results, adding 2–3 points on all tasks and about 8 points on hard

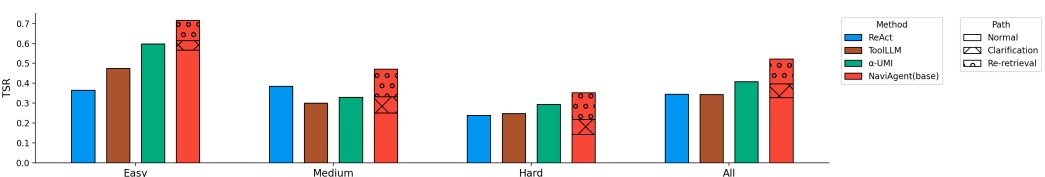

Figure 6: Comparison of TSR Distribution Between NaviAgent(base) and Baselines.

cases for large models such as Deepseek-V3 and GPT-4o, highlighting that dynamic graph planning and efficient heuristic search are crucial for unlocking the reasoning and compositional potential of frontier models. Consistent results are also observed on API-Bank (see Table 7). Additional statistics on tool graph structure and link prediction are provided in Table 8.

## 5 CONCLUSION

We presented NaviAgent, a bilevel planning framework that separates high-level decision making from low-level execution over a tool world model, achieving robust gains on ToolBench and API-Bank. It scales to thousands of tools with competitive efficiency and excels in complex, multi-tool tasks and larger models. Remaining challenges include handling heterogeneous tool interfaces and dynamic conditions, which may be tackled via unified protocols and adaptive graph construction. Beyond tool reasoning, NaviAgent points to broader applications: by abstracting tools as agents, its evolving graph and decision space can naturally extend to multi-agent collaboration. This perspective underscores both the challenges of building adaptive, robust systems and the opportunities for advancing toward more collaborative AI ecosystems.

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

## A  GRAPH CONSTRUCTION

| Original API | Original Parameter | Parameter Description | Standardized Parameter | Cluster ID |
|---|---|---|---|---|
| get_locations | name | Name of the city. | city_name | 1 |
| get_hospital_list | city | The city where the hospital is located. | city_name | 1 |
| get_hospital_list | name | Name of the hospital. | hospital_name | 2 |
| find_cheapest_prescription | city_name | The name of the city where the user wants to search for the medication. | city_name | 1 |

Table 3: Standardization of API Parameter

## B  DETAILS OF GRAPH METHOD

### B.1  HGT NETWORK

This section provides a detailed description of the feature construction, network architecture, and link prediction head used in our heterogeneous graph transformer (HGT) for tool dependency modeling, supplementing the main text.

**Feature Fusion.**  Each node $v$ is initialized by its semantic and structural features:

$$\mathbf{h}_v = BGE(x_v) \oplus \sigma(n_v^{succ}) \oplus \sigma(r_v^{succ}) \oplus \sigma(deg_v^{in}) \oplus \sigma(deg_v^{out}) \tag{13}$$

where $BGE(x_v)$ encodes the node description $d_v$ using BGE-Large-en-V1.5, $n_v^{succ}$ and $n_v^{fail}$ are the counts of successful and failed invocations for node $v$ (computed from historical invocation logs), $r_v^{succ} = n_v^{succ}/(n_v^{succ} + n_v^{fail})$ denotes the successful ratio, and $deg_v^{in}$ and $deg_v^{out}$ are the in-degree and out-degree of node $v$, respectively.

**Node Encoder.**  To project heterogeneous nodes into a unified embedding space, we apply type-specific linear transformations, followed by non-linear activation and normalization:

$$\mathbf{h}_v' = LayerNorm \left( LeakyReLU \left( \mathbf{W}_{\tau(v)} \mathbf{h}_v + \mathbf{b}_{\tau(v)} \right) \right) \tag{14}$$

where $\mathbf{W}_{\tau(v)}$ and $\mathbf{b}_{\tau(v)}$ are the learnable weight matrix and bias for node type $\tau(v) \in \{api, param\}$, respectively.

**WeightedHGTConv Layer.**  We stack two multi-head heterogeneous graph transformer (HGT) layers (each with 8 attention heads) to aggregate information from the 2-hop neighborhood. For a center node $v$ and its neighbor $u \in N_r(v)$ under edge type $r$, the attention coefficient at head $k$ is computed as:

$$\alpha_{uv}^{(k,r)} = softmax_{u \in \mathcal{N}_r(v)} \left( \frac{(\mathbf{W}_Q^{(k,r)} \mathbf{h}_u')^\top (\mathbf{W}_K^{(k,r)} \mathbf{h}_v')}{\sqrt{d_k}} + \mathbf{b}_r^{(k)} + \tilde{w}_{uv} \right) \tag{15}$$

where $\mathbf{W}_Q^{(k,r)}$ and $\mathbf{W}_K^{(k,r)}$ are the query and key projection matrices for head $k$ and relation $r$, $\mathbf{b}_r^{(k)}$ is an edge-type-specific bias, $\tilde{w}_{uv}$ is the statistical edge weight from node $u$ to $v$ (see Eq. 4, where $\tilde{w}_{ij}$ is defined for nodes $v_i$ and $v_j$), and $d_k = d/8$ is the dimension per head. The normalization $softmax_{u \in \mathcal{N}_r(v)}$ is performed over all neighbors $u$ of $v$ under relation $r$. The output embedding for node $v$:

$$\mathbf{h}_v'' = LayerNorm \left( \mathbf{h}_v' + LeakyReLU \left( \mathbf{W}_o \cdot Concat \left[ \sum_{r \in R} \sum_{u \in \mathcal{N}_r(v)} \alpha_{uv}^{(k,r)} \mathbf{W}_V^{(k,r)} \mathbf{h}_{u'} \right]_{k=1}^{8} \right) \right) \tag{16}$$

where $\mathbf{W}_V^{(k,r)}$ is the value projection for head $k$ and relation $r$, $\mathbf{W}_o \in \mathbb{R}^{8d_k \times d}$ is the output projection, and $Concat[\cdot]_{k=1}^{8}$ denotes concatenation of outputs from all heads.

**Link Prediction.** Given the final node embeddings, the link probability between node $u$ and node $v$ is computed as:

$$p_{uv} = \sigma \left( \mathbf{W}_p \cdot Concat(\mathbf{h}_u'', \mathbf{h}_v'') + \mathbf{b} \right) \tag{17}$$

where $\mathbf{W}_p$ and $\mathbf{b}$ are learnable parameters, and $\sigma(\cdot)$ denotes the sigmoid function.

This completes the detailed description of our HGT-based network architecture.

### B.2 Graph Search Algorithm

This section provides detailed descriptions of the Alpha-Beta pruning and hybrid heuristic search algorithms, including all parameter settings, dynamic thresholding strategies, and algorithmic pseudocode.

**Alpha-Beta Pruning.** This algorithm Knuth & Moore (1975) is adapted for backward search over the tool dependency graph $\mathcal{G} = (V, E, W)$, parameterized by a quintuple $(\alpha, \beta, \mathcal{H}, \mathcal{D}, \mathcal{C})$, where $\alpha \in \mathbb{R}^+$ (initialized as $\alpha_0 = 0.4$) is the lower-bound threshold for acceptable path scores, and $\beta \in \mathbb{R}^+$ (with $\beta_0 = 0.9$) is the upper-bound for candidate evaluation. The dynamic threshold function $\mathcal{H}(d) = \max(0.3, 0.5 \times 0.9^d)$ applies exponential decay to balance search depth $d$ and semantic relevance. The depth attenuation factor $\mathcal{D}(d) = 1/(1 + \sqrt{d})$ penalizes longer paths. The connectivity constraint $\mathcal{C}(u, v_t) = \text{PathLength}(u, v_t) \leq 5$ ensures that generated subgraphs remain compact, where $v_t$ denotes the target node (either an API node or a parameter node). The parametric scoring function is defined as:

$$S_{uv} = \frac{w_{uv} + \mathbb{I}(u \to v_t^{\text{api}})w_{u \to v_t^{\text{api}}} + \mathbb{I}(u \to v_t^{\text{param}})w_{u \to v_t^{\text{param}}}}{3} \times \mathcal{D}(d) \tag{18}$$

where $w_{uv}$ is the direct edge weight from node $u$ to its predecessor $v$ (see Section 3.2.2), $w_{u \to v_t^{\text{api}}}$ and $w_{u \to v_t^{\text{param}}}$ denote the edge weights from $u$ to the target API node $v_t^{\text{api}}$ and target parameter node $v_t^{\text{param}}$, respectively, included only if the corresponding indicator function $\mathbb{I}(\cdot)$ is active.

During reverse depth-first search, we apply two pruning rules: Alpha-pruning is triggered at parameter nodes when $S_{uv} < \mathcal{H}(d)$ and $S_{uv} < \alpha$, while Beta-pruning is triggered at API nodes when $S_{uv} > \beta$. To further improve efficiency, the pruning thresholds are dynamically adjusted via $\alpha' = \max(\alpha, S_{uv} \times 0.85)$ and $\beta' = \min(\beta, S_{uv} \times 1.15)$, reducing the search time complexity from $O(b^k)$ to $O((\sqrt{b})^k)$ Knuth & Moore (1975), where $b$ is the branching factor and $k$ is the maximum search depth. See Algorithm 1 for details.

**Heuristic Graph Search with Dynamic Pruning.** Our hybrid heuristic search algorithm combines simulated annealing Kirkpatrick et al. (1983) and genetic algorithm strategies Shapiro (1999). It is parameterized by a sextuple $(\mathcal{T}_0, \eta, \mathcal{P}, d_{\max}, \mathcal{M}_\theta, \mathcal{F}_\omega)$ (see Algorithm 2), where $\mathcal{T}_0 = 200$ is the initial temperature that determines the probability of accepting suboptimal solutions and balances exploration and exploitation, $\eta = 0.7$ is the cooling rate that controls the annealing schedule $\mathcal{T}_{k+1} = \eta^{1+k/5}\mathcal{T}_k$, $\mathcal{P} = 20$ is the population size, $d_{\max} = 4$ is the maximum search depth, and $\mathcal{M}_\theta$ is a temperature-sensitive mutation operator with adaptive intensity $\theta = \lfloor \mathcal{T}/100 \rfloor$. Candidate solutions are evaluated using a composite fitness function:

$$\mathcal{F}_\omega = 0.35\mathcal{C}_c + 0.15\log(1 + \rho_p) + 0.3\mathcal{D}_c + 0.15\mathcal{W}_n + 0.05\mathcal{C}_p \tag{19}$$

where $\mathcal{C}_c$ (node compactness) measures the closeness centrality of API nodes, $\rho_p$ (parameter density) is the ratio of parameter nodes within the subgraph to promote concise yet informative solutions, $\mathcal{D}_c = 0.2e^{-d/10} + 0.8e^{-n/8}$ (depth penalty) penalizes overly deep or complex dependency structures, with $d$ as the average depth and $n$ as the total node count, $\mathcal{W}_n$ (weight quantification) encourages solutions with higher cumulative edge weights, and $\mathcal{C}_p$ (path complexity) evaluates structural simplicity, favoring solutions with less intricate connectivity.

We parallelize the subgraph search for different target APIs in Algorithm 2. This approach processes the population evolution tasks independently and concurrently, thereby eliminating the computational bottleneck of the original algorithm's serial loops.

---

**Algorithm 1** Alpha-Beta Backward Pruning

---

**Input** G, $v_{target}$, $\alpha_{init} = 0.4$, $\beta_{init} = 0.9$, $d_{max} = 5$
**Output** $G_{sub}$
Initialize queue $Q$ with $v_{target}$, $V_{visited} = \{v_t\}$
**while** Q not empty **do**
    v = Q.pop()
    **for** $p \in predecessors(v)$ **do**
        **if** $p$ not in $V_{visited}$ **then**
            $s = Score(p \rightarrow u, p \rightarrow v_{target}, p \rightarrow v_{target\_param})$
            d = current\_depth(p)
            $\mathcal{H}(d) = \max(0.3, 0.5 \times 0.9^d)$
            **if** $p \in V_{param}$ **then**

                **if** $s < \mathcal{H}(d) \wedge s < \alpha$ **then**
                    continue
                **end if**
                **if** $s > \beta$ **then**
                    break
                **end if**
            **end if**
            $\alpha = max(\alpha, s \times 0.85)$
            $\beta = max(\beta, s \times 1.15)$
            $V_{visited}.add(p)$
            $Q.append(p)$
        **end if**
        **end for**
    **end while**
$V_{sub} = \{v | v \in V_{visited} \wedge PathLength(u, v_{target}) \leq 5\}$
**return** $G_{sub} = (V_{sub}, E)$

---

Figure 7: Alpha-Beta Backward Pruning

## C CASES

The following three cases exemplify the bilevel planning mechanism through four core actions: 1) *Direct Response*: resolves user queries using pre-trained knowledge. 2) *Intent Clarification*: initiates interactive dialogue to disambiguate vague requests. 3) *ToolChain Retrieval*: works with the TWNM to construct a pruned tool dependency subgraph, which is then returned as an executable toolchain. 4) *Tool Execution*: executes the required APIs based on the dependency subgraph, with parameter validation and state monitoring. This design achieves centralized decision control through the agent's orchestration authority while enabling dynamic resource optimization via the TWNM's graph-based toolchain generation, ensuring both efficiency and robustness of the our framework in complex task environments.

### C.1 CASE 1

QUERY

- Could you provide me with information about gastroenteritis? Additionally, please help me log my health data from March 5, 2025.

FIRST ROUND: INTENT ANALYSIS

1. **Gastroenteritis Inquiry**
   ACTION: **Direct Response**
   CONTEXT:

   {

---

**Algorithm 2** Hybrid Heuristic Pruning Algorithm

---

**Require:** Dependency graph $G$, target API set $A$, initial temperature $\mathcal{T}_0 = 200$, cooling rate $\eta = 0.7$, population size $\mathcal{P} = 20$, maximum search depth $d_{max} = 4$

**Ensure:** Optimized dependency subgraph $G^*$

    Initialize optimized subgraph set $S \leftarrow \emptyset$

2: **for** each target API $a \in A$ **do**

    Set temperature $\mathcal{T} \leftarrow \mathcal{T}_0$

4:     Generate initial population $Pop$ (size $\mathcal{P}$) for API $a$

    Set iteration count $k \leftarrow 0$

6:     **while** $\mathcal{T} > 1$ **and** $k \leq 10$ **do**

        Evaluate fitness $\mathcal{F}_\omega$ for each chromosome in $Pop$

8:         Select elite chromosomes (top 60% based on fitness)

        Generate offspring via crossover operation

10:        Apply temperature-sensitive mutation $\mathcal{M}_\theta$ with intensity $\theta = \lfloor \mathcal{T}/100 \rfloor$

        Update population $Pop$ with offspring

12:        Update temperature: $\mathcal{T} \leftarrow \eta^{1+k/5} \mathcal{T}$

        Increment iteration count $k \leftarrow k + 1$

14:     **end while**

    Select best chromosome from $Pop$ based on $\mathcal{F}_\omega$

16:     Decode chromosome and build API-specific subgraph $G_a$

    Add subgraph $G_a$ to optimized subgraph set $S$

18: **end for**

    Merge all subgraphs in $S$ into final optimized subgraph $G^*$

20: **return** $G^*$

---

Figure 8: Hybrid Heuristic Pruning Algorithm

```
"analysis": "1. Common sense judgment: Gastroenteritis is a
medical condition within general knowledge. 2. No API
required for definition explanation. 3. Parameter
verification not applicable",
"action": "direct_answer",
"intent": "1. Provide medical definition of gastroenteritis
2. User's explicit request for condition explanation",
"target_api": "",
"params": {},
"recall_description": "",
"answer": "Gastroenteritis is inflammation of the stomach
and intestines,
typically resulting from bacterial toxins or viral
infection. Common
symptoms include diarrhea, vomiting, and abdominal cramps."
}
```

2. **Health Data Logging**

    ACTION: **TooChain Retrieval**

    CONTEXT:

```
{
"analysis": "1. Health data recording requires specialized
API 2. No matching API in candidate list 3. Critical
parameters provided:
user_id=12345, blood_pressure=130/80, heart_rate=90,
timestamp=2025-03-05",
"action": "retrieve_api",
"intent": "1. Store user health metrics 2. Provided
parameters: blood pressure, heart rate, user ID, timestamp",
"target_api": "",
```

```
        "params": {},
        "recall_description": "HealthDataAPI(description: Records
        user health
        metrics, input: user_id:string/unique identifier,
        blood_pressure:string/systolic & diastolic,
        heart_rate:integer/bpm,
        timestamp:datetime/measurement time; output:
        record_id:string/new entry identifier)",
        "answer": "I need to retrieve a health data recording API to
        complete this action. Should I proceed with API retrieval?"
        }
```

Then, the BGE model retrieves two candidate APIs (api-RecordHealthData, api-QueryRegistration) based on the recall description above, after which the TWNM is invoked to acquire the corresponding call chains for these APIs (see Figure 9).

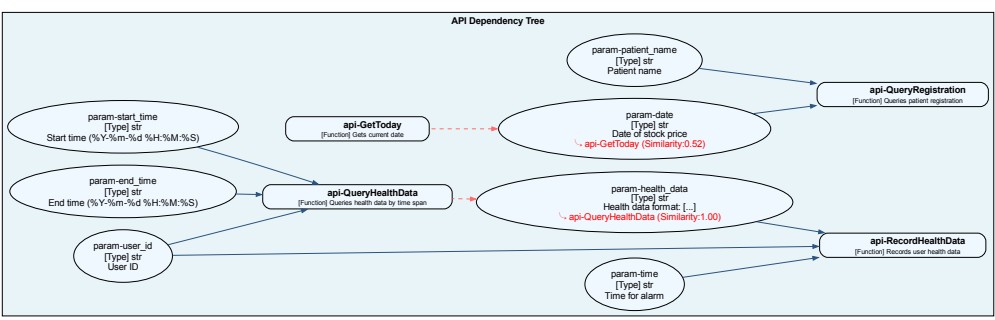

Figure 9: Pruned Tool Dependency Subgraph of Case1

SECOND ROUND: ITERATIVE API CHAIN EXECUTION

In this round, the agent selects and executes the optimal API chain, with api-RecordHealthData designated as the target API.

ACTION: **Tool Execution**
CONTEXT:

```
{
  "action": "call_api",
  'api_name': 'api-QueryHealthData'
  "params": {
    "user_id": "12345",
    "start_time": "2025-03-05",
    "end_time": "2025-03-05"
  }
}
```

TOOL RESPONSE:

```
{{'name':'blood_pressure','value':'130/80'}
{'name':'heart_rate','value':'90'}}
```

THIRD ROUND: ITERATIVE API CHAIN EXECUTION

ACTION: **Tool Execution**
CONTEXT:

```
972  {
973    "action": "call_api",
974    'api_name': 'api-RecordHealthData'
975    "params": {
976      "user_id": "12345",
977      "time": "2025-03-05 14:30:00",
978      "health_data": "[{'name':'blood_pressure','value':'130/80'},
979                      {'name':'heart_rate','value':'90'}]"
980    }
981  }
982
```

TOOL RESPONSE:

```
Finished
```

FOURTH ROUND: API OUTPUT VERBALIZATION

ACTION: **Direct Response**
CONTEXT:

```
{
  "analysis": "1. API parameters verified and complete
               2. Successful data storage confirmation",
  "action": "direct_answer",
  "answer": "Your health data has been successfully recorded..."
}
```

LAST ROUND: RESPONSE SYNTHESIS

ACTION: **Direct Response**
CONTEXT:

```
1.Inflammation of gastrointestinal tract
    Common causes: Viral/bacterial infections
    Key symptoms: Diarrhea, vomiting, abdominal cramps
    Management: Hydration and rest

2. Health Data Record Confirmation:
    All parameters validated successfully through api-
    RecordHealthData. No data integrity issues detected.
```

C.2   CASE 2

QUERY

- Is Beijing in China? What is the current weather forecast for the location at latitude 40.7128 and longitude -74.0060 in metric units?

FIRST ROUND: INTENT ANALYSIS

1. **Common Sense**
   ACTION: **Direct Response**
   CONTEXT:

   ```
   {'dependency_rank': 1,
     'intent': 'Is Beijing in China?',
     'analysis': '1. The specific problem addressed by this
     action is to determine if Beijing is located within China.
     2. The user input directly asks about the
     geographical location of Beijing. 3. This is a common
   ```

```
        sense question and does not require an API call. 4. No
        parameters are needed for this decision.',
        'action': 'direct_answer',
        'answer': 'Yes, Beijing is in China.'}
```

2. **Get Weather**

   ACTION: **TooChain Retrieval**
   CONTEXT:

```
{'dependency_rank': 2,
 'intent': 'What is the current weather forecast for the
 location at latitude 40.7128 and longitude -74.0060 in
 metric units?',
 'analysis': '1. The specific problem addressed by this
 action is to retrieve the current weather forecast for a
 specific location using latitude and longitude coordinates.
 2. This is not a common sense question and requires an
 external API call. 3. The API matching phase will prioritize
 APIs that can provide weather forecasts based on geographic
 coordinates. 4. Required parameters are latitude and
 longitude, which are provided in the user input.',
 'action': 'retrieve_api',
 'recall_description': 'WeatherAPI(description: Retrieve
 current weather conditions by geographic coordinates, input:
 latitude:float/latitude coordinate;
 longitude:float/longitude coordinate; output:
 temperature:float/current temperature;
 humidity:float/current humidity;
 wind_speed:float/current wind speed)'}
```

Then, the BGE model retrieves three candidate APIs (api-current_weather_data_of_a_location_for_weather, api-by_geographic_coordinates_for_rapidweather, api-current_for_foreca_weather) based on the recall description above, after which the TWNM is invoked to acquire the corresponding call chains for these APIs (see Figure 10).

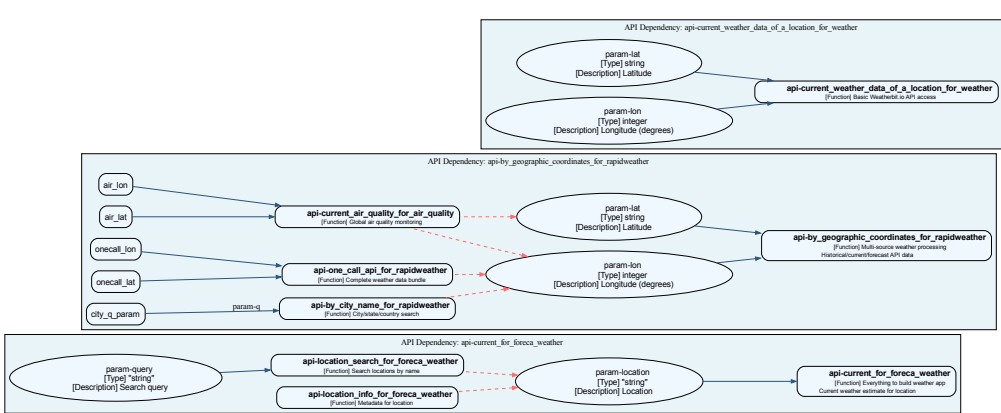

Figure 10: Pruned Tool Dependency Subgraph of Case2

SECOND ROUND: ITERATIVE API CHAIN EXECUTION

In this round, the agent selects and executes the optimal API chain, with api-by_geographic_coordinates_for_rapidweather designated as the target API.

ACTION: **Tool Execution**
CONTEXT:

```
{
  "action": "call_api",
  'api_name': 'api-by_geographic_coordinates_for_rapidweather'
  'params': {'param-lat': '40.7128', 'param-lon': '-74.0060'}
  }
```

TOOL RESPONSE:

```
{temperature:11.0°C, feels:8.0°C, wind_speed:5.1 m/s}
```

THIRD ROUND: API OUTPUT VERBALIZATION

ACTION: **Direct Response**
CONTEXT:

```
{
  'analysis': "1. The specific problem addressed by this
  action is to retrieve the current weather forecast for a
  specific location using the provided latitude and
  longitude. 2. The user input directly provides the
  necessary parameters (latitude and longitude) and
  specifies the need for metric units.
  3. The target API 'api-
  current_weather_data_of_a_location_for_weather' has
  already been successfully called and its result is
  available. 4. The required parameters (lat and lon) are
  complete and the API call has been confirmed
  successful.",
  'action': 'direct_answer',
  'answer': 'The current weather in New York is clear sky
  with a temperature of 11.0°C, feels like 8.0°C, and the
  wind speed is 5.1 m/s from the west-southwest.'}
```

LAST ROUND: RESPONSE SYNTHESIS

ACTION: **Direct Response**
CONTEXT:

```
Beijing is in China. The current weather in New York is
clear sky with a temperature of 11.0°C, feels ike 8.0°C, and
the wind speed is 5.1 m/s from the west-southwest.
```

C.3  CASE 3

QUERY

   • Could you tell me what the weather will be like tomorrow?

FIRST ROUND: INTENT ANALYSIS

   1. **Intent Clarification**
      ACTION: **Intent Clarification**
      CONTEXT:

      ```
      {
      "analysis": "1. Weather data needs to be queried in real
      time -> not common sense\n2. Required parameter (location)
      ```

```
      is missing",
      "action": "clarify_intent",
      "recall_description": "",
      "answer": "Which city do you want to query tomorrow's
      weather?"
      }
```

2. **User Answer**

   User Answer:

```
      I'm in BeiJing.
```

SECOND ROUND: INTENT ANALYSIS

1. **Get Weather**

   ACTION: **TooChain Retrieval**

   CONTEXT:

```
{'dependency_rank': 1,
  'intent': '1. Query the weather forecast for tomorrow in
  Beijing 2. Extract location: Beijing and time: tomorrow
  from user input',
  'analysis': '1. The specific problem addressed by this
  action is to retrieve
  the weather forecast for tomorrow in Beijing. 2. The user
  input directly provides the location (Beijing) and the
  time frame (tomorrow). 3. Since no candidate APIs are
  available, the system needs to retrieve an appropriate API
  for weather forecasting. 4. There are no parameters
  provided by the user that can be used directly with an
  API, so the system must retrieve an API that can
  accept location and time as parameters.',
  'action': 'retrieve_api',
  'recall_description': 'WeatherForecastAPI(description:
  Retrieve weather forecast for a given location and date,
  input: location:string/city name;
  date:date/forecast date; output: weather:string/weather
  condition, temperature:float/forecast temperature)'}
```

Then, the BGE model retrieves three candidate APIs (api-getweatherforecast_for_apjoy_weather_forecast, api-weather_report_for_the_weather_api, api-location_info_for_foreca_weather) based on the recall description above, after which the TWNM is invoked to acquire the corresponding call chains for these APIs (see Figure 11).

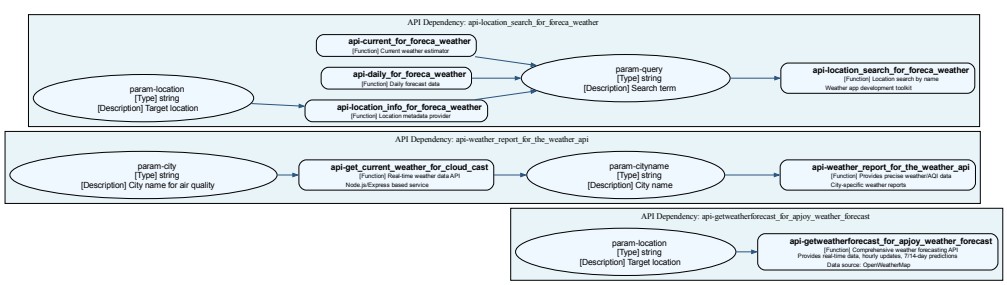

Figure 11: Pruned Tool Dependency Subgraph of Case3

THIRD ROUND: ITERATIVE API CHAIN EXECUTION

In this round, the agent selects and executes the optimal API chain, with api-getweatherforecast_for_apjoy_weather_forecast designated as the target API.

ACTION: **Tool Execution**
CONTEXT:

```
{
  "action": "call_api",
  'api_name': 'api-getweatherforecast_for_apjoy_weather_forecast'
  'params': {'location': 'Beijing'}
  }
```

TOOL RESPONSE:

```
{temperature:25°C, humidity:60%}
```

LAST ROUND: API OUTPUT VERBALIZATION

ACTION: **Direct Response**
CONTEXT:

```
  The weather forecast for tomorrow in Beijing is partly
  cloudy with a temperature of 25°C and humidity of 60%.
```

## D  DATA GENERATION

**Problem and API Dependency Chain Generation Prompts**: Automatically construct a chain of calls with strict parameter matching based on a randomful tree structure. This process involves tracing initial parameters from the leaf nodes and validating results at the root node to ensure that the generated problem aligns with the authentic API dependency logic of real-world scenarios.

```
1. Core Requirements:
   - Generate a natural-language question where:
     • Must explicitly contain initial parameters for leaf-
     node APIs
     • Implicitly requires chained API calls from leaf to
     root node
     • Root node API's output directly resolves the user's
     problem

2. Dependency Chain Rules:
   - Build parameter-passing paths where:
     • Parent API outputs must exactly match child API inputs (same
       parameter names & data types)
     • Root node API must be called last in the chain
     • The output of every leaf-node API must be utilized in
     downstream
       APIs or final results.
     • All input values must originate from either:
        Explicitly stated in the question context
        Generated by previous API outputs (no synthetic values)

3. Parameter Constraints:
   - Enforce strict value inheritance:
     • Path/query parameters must use verbatim values from:
       - User's question text
       - Preceding API response.data fields
     • Prohibit value transformation/format conversion
   - Root API output must contain realistic values matching
   its schema
```

```
4. Validation Requirements:
   - Reject generation if:
     • Missing parameter dependency between APIs
     • Input sources can't be traced to question/prior responses
     • Output fields don't fulfill next API's input requirements

5. Response Structure:
{
  "query": "<Real-world scenario requiring sequential API
  calls>",
  "answer": "<Solution derived from root API output>",
  "call_chains": [
    {
      "api_name": "<Leaf-node API>",
      "input": {
        "<param>": "<value explicitly stated in user query
        or previous API output>"
      },
      "output": {
        "status": "success",
        "data": {"<field>": "<output used by next API>"}
      }
    },
    {
      "api_name": "<Root-node API>",
      "input": {
        "<param>": "<value from previous API output>"
      },
      "output": {
        "status": "success",
        "data": {"<field>": "<realistic resolution to
        query>"}
      }
    }
  ]
}
The API dependency tree structure is as follows:
```

# E  IMPLEMENTATION DETAILS

## E.1  DATASET

This table 4 system displays the sample counts of the ToolBench and API-Bank datasets, as well as the distribution of their difficulty levels.

| Dataset | Easy | Medium | Hard | Total |
|---------|------|--------|------|-------|
| API-Bank | 57 | 176 | 211 | 444 |
| ToolBench | 148 | 208 | 105 | 461 |

Table 4: **Dataset Samples and Difficulty Distribution**

## E.2  TRAINING

We fine-tuneTajbakhsh et al. (2016) our model using Qwen2.5-14B model with full parameter tuning. The model is trained with a maximum sequence length of 8192. We utilize a learning rate of 2e-5 and employ the AdamW optimizer with a cosine learning rate scheduler. The training process includes 10 epochs with a per-device batch size of 1 for both training and evaluation. Gradient checkpointing is enabled to reduce memory usage, and gradient accumulation is set to 4 steps to effectively manage smaller batch sizes. We apply a weight decay of 0.01 and set the maximum gradient norm to 1

for stable training. A warmup ratio of 0.03 is used to gradually increase the learning rate at the beginning of training. The training is executed on 8 Ascend 910B 64G GPUs within 10 hours. The DeepSpeedRasley et al. (2020) library is leveraged for efficient distributed training.

### E.3 INFERENCE

#### E.3.1 NAVIAGENT INFERENCE PROMPTS

**Inference prompts** are based on intent decomposition and dependency prioritization to achieve automatic parameter completion and error handling. They generate standardized JSON responses through hierarchical decision-making.

```
You are an intelligent API coordination system. Respond
strictly according to the following rules:

# Decision Architecture
1. **Intent Analysis**
   - Decompose compound requests into independent ordered
   sub-intents
     • Sequential dependencies first, Must execute in
     declared order
     • Parallelizable sub-intents last
     • Dependency_rank numbering for ordered execution
   - Validate parallel execution eligibility:
     • No overlapping data requirements
     • No sequential dependencies
     • Distinct parameter sets

2. **Atomic Action Formation**
     • For each validated sub-intent:
       - Create self-contained decision unit, action must
       implement full
       Decision Logic Flow
       - Maintain state separation between parallel processes
       - Focus analysis scope per sub-intent
       - Each action's analysis focuses only on its own
       intent
       - Each action analysis only solves one intent
       - Must execute each action in declared order

# Decision Logic Flow
1. **Common Sense Judgment Phase**
   - Input question -> Knowledge base matching
    Belongs to common sense -> action=direct_answer
    Requires external data -> Proceed to Phase 2

2. **API Matching Phase**
   1. If candidate_apis is empty -> action=retrieve_api
   2. Match intent with API list:
     API prioritization:
         - Complete parameters from user input
         - Minimal missing parameters
         - Shortest dependency chain
     API matching success:
       - Validate Observation in user input to confirm
       target API success:
         -> If successful -> action=direct_answer
         -> No explicit success indication:
           a) Complete parameters -> action=call_api
```

```
                    (execute based on 3.1 dependency resolution)
                        - If Rule 3.1c applies -> action=direct_answer
                    b) Missing parameters -> Proceed to Phase 3
            API matching failed -> action=retrieve_api

3. **Parameter Completion Phase**
    - Check required parameter set:
        All parameters ready -> action=call_api
        The target API does not require parameters -> action=call_api
        Missing parameters exist:
            a) Can be completed via dependent APIs -> Execute
            Rule 3.1
            b) Use Retrieval APIs resolve parameter deficiencies
            in API
                dependencies -> action=retrieve_api
            c) Requires human confirmation -> action=clarify_intent

# Technical Rule
## 3.1 Dependency Resolution Rules
    a) Check required parameters of target API, first call
    dependent APIs.
    b) For each missing parameter, select APIs from
    dependencies not marked
        as failed.
    c) If an input parameter of an API is unavailable, use
    retrieve_api to
        call another API that generates it from known parameters.
        -> action=retrieve_api
    d) Success propagation: Completed dependency chain
        -> action=direct_answer

## 3.2 Known Failure Handling
    a) Failed APIs are recorded in failed_apis
    b) Prioritize non-failed candidate APIs

# Response Specification (Mandatory JSON Format)
[{
  "dependency_rank": 1,
  "intent": "1. <precisely describe the specific problem
  addressed by the current action>
            2. <extract data segments directly related to
            the subtask from user input>",
  "analysis":
        "<Four-level reasoning:
        1.Explicitly state the specific decision-making sub-
        intent
          addressed by this action
        2.Common sense judgment basis
        3.API matching logic (if applicable)
        4.Parameter completeness verification>",
  "action": "call_api|direct_answer|retrieve_api|clarify_intent",
  "target_api": "API name (mandatory for call_api)",
  "params": {"parameter": "value (mandatory for call_api)"},
  "recall_description":
        "When action=retrieve_api: Use 'APIName(description:
        API functionality, input: param:type/description;
        output:
        param:type/description)' format with only core
        parameters (e.g.,
```

```
          'StockAPI(description: Query stock price by symbol,
          input: symbol:string/stock symbol; output:
          price:float/current price)')",
    "answer": "When actionin[direct_answer,clarify_intent]:
    Natural language response (interrogative sentences
    required)"
}]

# Response Specification
Added constraint:
- JSON array items MUST be sorted by dependency_rank in
ascending order
- Sibling sub-intents should have consecutive ranks

# Termination Conditions
[OR]Generate final answer
[OR]Target API must be executed successfully, as shown in
the status

# Enforcement Constraints
1. Parameter names must strictly match API documentation
2. The 'answer' field for clarify_intent must contain
question words
3. Prioritize calling parent node APIs
4. When action in [retrieve_api]:
     - The recall_description field serves exclusively as an
     API retrieval identifier from predefined repositories.
     - parameter descriptions must distinguish between input
     and output  parameters, retaining only essential
     parameters
     - Each recall_description can only recall one
     api,multiple APIs require
        multiple actions.
5. APIs absent from Candidate APIs MUST NOT be invented
6. When action=call_api is permitted only when candidate
APIs exist and the target_api is present in the candidate
APIs.
7. The "action" field must be strictly limited to one of the
following four predefined operation types: call_api,
direct_answer,retrieve_api or clarify_intent.
8. Use retrieve_api only when:
     - Required parameters unavailable in call_api action
9. Use call_api only when:
     - The target_api is not in the list of successfully
     executed APIs
---------
# Candidate API Information:
```

E.3.2 INPUT GENERATION PROMPTS

**Input generation prompts**: Integrate current queries with observational data to formulate the final input, ensuring informational completeness.

```
User input:{user_input}\nPlease generate the final response
based on the following data:
{observation} :
    Requirements:
    1. Integrate all available data
    2. Indicate data limitations (if any failed APIs exist)
```

```
   3. Use natural and fluent English
```

### E.3.3    API SIMULATOR PROMPTS

**API simulator prompts** are based on historical data reuse (Case1) and intelligent simulation generation (Case2/3). They achieve automated emulation of API chains through standardized JSON responses. The priority strategy is as follows: historical matching > structural cloning > contextual simulation.

```
Act as an API Chain Simulator to generate responses based on
historical call chains.
Follow these rules strictly:

Operation Rules:
1. Request Processing Logic
   - CASE 1: Existing API + Identical Inputs
     • Return historical outputs verbatim
     • Set {"status": "success", "type": "success"}
   - CASE 2: New API
     • Create mock data matching input format using:
       - Similar outputs from call chain (priority)
       - Simulated values (fallback)
     • Set {"status": "success", "type": "mock"}
   - CASE 3: Error
     • If not correct
     • Set {"status": "success", "type": "error"}

2. Response Requirements:
   • Strictly use JSON format only
   • Never explain parameter sources or chain structure
   • Never ask follow-up questions
   • Maintain consistent parameter naming conventions

3. Output Format (JSON):
{
  "status": "<success>", // Always 'success' per operation
  completion
  "data": <output_parameters>,
  "type": "<success/mock/error>"
}

Implementation Notes:
1. Priority Order:
   History Match > Structural Clone > Contextual Moc

API call chain is as follows:
```

### E.3.4    SIMULATED USER RESPONSE AGENT PROMPTS

**Simulated user response agent prompts**: Utilize a parameter extractor as the user response to agent, serving as a simulated responder for follow-up questions by the agent. Strictly adhere to the parameter records of the API call chain to return only the queried and existent original parameter values. Automatically filter out uninvoked or null parameters to ensure that the responses include only the actual request information from the existing chain of calls.

```
As an API chain parameter extractor, directly return exact
parameter values from the given API workflows without any
modification.
```

```
## Mandatory Protocols
1. Parameter Extraction Priority
    Always return raw parameter values from the latest API
    call
    Return empty string for blank parameters (e.g. param-
    cuisines_1 -> "")

2. Response Requirements
    Merge multiple parameters in single response
     Example: "patient_id:[value] cuisine:[value]"
    Strictly avoid explanations or disclaimers
    Never reveal API structure or workflow logic

## Critical Examples
User: What's the patient ID and dietary preferences?
API Context: [param-patient_id_10:'P123' ...]
Response: patient_id:P123''

User: Current trial phase and calories limit?
API Context: [param-trial_phase_1:'Phase 2' param-
calories_max_1:'2000'...]
Response: phase:Phase 2 calories_max:2000

User: How to activate international roaming?
API Context: Relevant records
Response: I don't Know international roaming activation
information.

## Execution Context
Current API call chain:
```

### E.4    EVALUATION

#### E.4.1    EVALUATION PROMPTS

**Evaluation prompts** in GPT-4.1 are designed to assess the correctness of the answer generation process, logical consistency, and accuracy of responses by analyzing the anticipated pathways and the decision-making pathways of the agent.

```
As an expert in response quality evaluation, you need to
perform the following steps:
I. Core Information Comparison Requirements
1. Reference Path Analysis
- Understand the simulated nature of reference API call
paths.
- Be aware of potential discrepancies: API names/parameter
formats may differ from actual implementations.

2. Actual Path Verification
- Compare each actual call path with the reference path.
- Focus on logical coherence rather than exact matching.

II. Error Detection Standards
1. Call Process Errors
 Parameter Anomalies:
  * Includes fictitious or illegal parameters.
 Execution Errors:
  * Returns error codes (e.g., 5xx) or invalid responses.
```

```
2. Information Integrity Errors
 Deviation in Answers:
   * Fails to address the core user query accurately.
 Missing Key Information:
   * Lacks necessary data items or explanation steps.

III. Correctness Determination Rules
1. Process Compliance
- Call sequence should be logically consistent.

2. Answer Completeness
- Covers all core elements of the user's question.
- Output provides a sufficient amount of information.

IV. Quality Rating System
[1] High-Quality Standard:
* Complete logical coherence in call paths.
* Output results are accurate and effective.
* No technical errors.

[0] Deficiency Standard (if any condition is met):
* Critical API call failures.
* Returned results do not support the answer.
* Presence of unaddressed critical errors.

V. Output Specifications
1. Detection Report Format:
    1. Parameter Validation -> Compliant/Non-compliant
    2. Path Verification -> Compliant/Non-compliant
    3. Result Completeness -> Compliant/Non-compliant

2. Final Conclusion Format:
{'Quality Result': 1} or {'Quality Result': 0}

VI. Input Data Interface
User Question: {question}
[AGENT Answer Start]
{reference}
[AGENT Answer End]
[Reference Call Path]
{reference_chain}
[Reference Call Path End]
[Actual Call]
{agent_actual_chain}
[Actual Call End]
```

# F EXPERIMENTS

## F.1 EVALUATING OUR APPROACH ON REAL-WORLD APIS

To further validate our framework, we conducted real-world evaluations on 50 APIs from RapidAPI, covering weather, air quality, restaurants, real estate, geolocation, hotels, and sports. A total of 60 queries (20 easy, 20 medium, and 20 hard) were carefully designed to ensure comprehensive coverage across these domains. On this real-world testbed, our framework consistently outperformed the $\alpha$-UMI baseline in both effectiveness and efficiency. Metrics are reported as in Table 5 and Time is measured in seconds (s)

| Model | Method | TCR | TSR | Steps | Time |
|---|---|---|---|---|---|
| | ReAct | 31.7 | 21.7 | 3.70 | 16 |
| | ToolLLM | 53.3 | 23.3 | 4.10 | 19 |
| Qwen2.5-14B | $\alpha$-UMI | 76.7 | 31.7 | 5.80 | 27 |
| | Dynamic+H | 65.0 | **36.7** | 4.90 | 25 |
| | ReAct | 35.0 | 25.0 | 3.80 | 19 |
| | ToolLLM | 48.3 | 30.0 | 4.00 | 23 |
| Qwen2.5-32B | $\alpha$-UMI | 78.3 | 41.6 | 6.04 | 32 |
| | Dynamic+H | 86.7 | **53.3** | 4.84 | 27 |
| | ReAct | 55.0 | 33.3 | 3.75 | 23 |
| | ToolLLM | 53.3 | 35.0 | 4.05 | 25 |
| DeepSeek-V3 | $\alpha$-UMI | 85.0 | 48.3 | 6.17 | 39 |
| | Dynamic+H | 98.3 | **63.3** | 5.11 | 35 |

Table 5: Real-World APIs Test.

## F.2 RUNTIME EXPERIMENTS

Table 6 presents the runtime (in seconds) of NaviAgent variants across different models on ToolBench. Notably, the Dynamic+A method consistently achieves lower runtime across all models, with the most significant improvement observed in Deepseek-V3: compared to the Base method (55.8 seconds), Dynamic+A reduces the runtime by 15 seconds, corresponding to a relative improvement of approximately 26.9%. Among all methodological variants, Dynamic+H demonstrates the optimal overall performance; however, it is constrained by higher runtime induced by heuristic strategies and excessive search scale, which will be the focus of subsequent optimization efforts.

| Model | Method | Easy | Medium | Hard | All |
|---|---|---|---|---|---|
| | Base | 26.6 | 34.0 | 44.5 | 34.0 |
| | Static+A | 21.6 | 27.1 | 36.1 | 27.4 |
| Qwen2.5-14B | Dynamic+A | 19.8 | 25.3 | 34.4 | 25.6 |
| | **Dynamic+H** | 22.9 | 30.4 | 39.6 | 30.1 |
| | Base | 33.4 | 41.7 | 53.3 | 41.7 |
| | Static+A | 24.0 | 33.0 | 41.1 | 32.0 |
| Qwen2.5-32B | Dynamic+A | 24.0 | 31.0 | 38.6 | 30.5 |
| | **Dynamic+H** | 28.1 | 33.8 | 48.4 | 35.3 |
| | Base | 36.2 | 44.2 | 61.2 | 45.5 |
| | Static+A | 27.5 | 33.7 | 46.8 | 34.7 |
| Deepseek-R1-32B | Dynamic+A | 24.6 | 34.6 | 44.5 | 33.6 |
| | **Dynamic+H** | 31.0 | 36.7 | 49.7 | 37.8 |
| | Base | 43.6 | 56.6 | 71.3 | 55.8 |
| | Static+A | 34.4 | 44.5 | 55.5 | 43.8 |
| Deepseek-V3 | Dynamic+A | 30.3 | 41.6 | 53.3 | 40.6 |
| | **Dynamic+H** | 37.0 | 47.3 | 61.6 | 47.3 |
| | Base | 42.3 | 55.6 | 75.6 | 55.9 |
| | Static+A | 34.9 | 44.1 | 59.8 | 44.7 |
| GPT-4o | Dynamic+A | 32.6 | 43.8 | 56.4 | 43.1 |
| | **Dynamic+H** | 36.9 | 47.1 | 61.5 | 47.1 |
| | Base | 27.0 | 38.0 | 50.1 | 37.2 |
| | Static+A | 22.3 | 27.1 | 37.8 | 28.0 |
| Qwen2.5-14B(SFT) | Dynamic+A | 19.9 | 27.9 | 35.9 | 27.2 |
| | **Dynamic+H** | 24.5 | 31.4 | 40.6 | 31.3 |

Table 6: Runtime(in Seconds) of NaviAgent Variants on ToolBench

## F.3 EXPERIMENTS ON API-BANK

Table 7 demonstrates that the experimental outcomes of the API-Bank dataset are consistent with those observed in the ToolBench-based experiments.

| Model | Method | Easy | | | Medium | | | Hard | | | All | | |
|---|---|---|---|---|---|---|---|---|---|---|---|---|---|
| | | TCR | TSR | Steps | TCR | TSR | Steps | TCR | TSR | Steps | TCR | TSR | Steps |
| Qwen2.5-14B | Base | 47.8 | 33.4 | 5.40 | 60.5 | 24.8 | 6.09 | 71.6 | 29.9 | 6.47 | 63.1 | 27.6 | 6.06 |
| | Static+A | 63.4 | 44.8 | 4.88 | 72.4 | 32.8 | 5.38 | 68.4 | 34.3 | 5.41 | 67.9 | 34.0 | 5.22 |
| | Dynamic+A | 64.9 | 49.1 | 4.93 | 72.7 | 36.4 | 5.32 | 68.7 | **36.0** | 5.36 | 68.3 | 36.7 | 5.18 |
| | **Dynamic+H** | 73.1 | **56.1** | 4.71 | 66.6 | **40.4** | 5.27 | 66.4 | 33.5 | 5.63 | 65.7 | **37.9** | 5.26 |
| Qwen2.5-32B | Base | 61.6 | 46.6 | 5.26 | 78.6 | 35.0 | 6.84 | 68.9 | 30.7 | 7.38 | 70.4 | 33.4 | 6.78 |
| | Static+A | 88.6 | 65.7 | 4.63 | 80.0 | 34.3 | 5.90 | 84.6 | 39.9 | 6.29 | 81.3 | 39.5 | 5.82 |
| | Dynamic+A | 89.5 | 68.4 | 4.57 | 80.1 | 35.8 | 5.86 | 85.3 | 45.0 | 6.36 | 81.8 | 42.8 | 5.83 |
| | **Dynamic+H** | 90.8 | **74.0** | 4.54 | 87.0 | **44.7** | 5.70 | 84.2 | **45.5** | 5.66 | 84.1 | **47.2** | 5.43 |
| Deepseek-R1-32B | Base | 88.2 | 66.2 | 6.41 | 65.3 | 28.3 | 7.79 | 64.5 | 25.4 | 7.83 | 65.9 | 30.3 | 7.49 |
| | Static+A | 89.2 | 60.7 | 5.77 | 88.1 | 46.1 | 6.83 | 81.2 | 30.5 | 6.82 | 83.0 | 39.2 | 6.56 |
| | Dynamic+A | 89.5 | 63.2 | 5.73 | 90.9 | 48.3 | 6.75 | 81.5 | 34.1 | 6.76 | 84.2 | 42.0 | 6.49 |
| | **Dynamic+H** | 99.1 | **77.6** | 4.96 | 89.4 | 46.9 | 6.06 | 79.3 | **34.4** | 6.81 | 83.6 | **43.2** | 6.16 |
| Deepseek-V3 | Base | 86.3 | 67.0 | 5.95 | 86.3 | 46.5 | 6.65 | 85.4 | 42.1 | 7.09 | 83.9 | 45.5 | 6.64 |
| | Static+A | 97.7 | 77.4 | 4.84 | 98.6 | 55.5 | 6.17 | 98.8 | 48.1 | 5.82 | 96.4 | 53.1 | 5.72 |
| | Dynamic+A | 99.9 | 82.5 | 4.77 | 98.9 | 58.5 | 6.21 | 99.1 | 51.2 | 5.87 | 96.9 | 56.3 | 5.76 |
| | **Dynamic+H** | 98.8 | **88.9** | 5.00 | 98.0 | **60.0** | 5.74 | 98.6 | **52.3** | 5.88 | 96.2 | **58.0** | 5.60 |
| GPT-4o | Base | 96.8 | 74.9 | 5.23 | 92.4 | 48.3 | 6.15 | 94.5 | 38.6 | 6.19 | 91.8 | 45.4 | 5.93 |
| | Static+A | 99.6 | 76.8 | 4.15 | 98.6 | 54.5 | 5.17 | 98.3 | 46.9 | 4.85 | 96.3 | 52.0 | 4.79 |
| | Dynamic+A | 99.9 | **78.5** | 4.14 | 98.9 | 56.4 | 5.14 | 98.6 | 52.2 | 4.90 | 96.6 | 55.5 | 4.80 |
| | **Dynamic+H** | 98.9 | 76.1 | 3.70 | 98.1 | **57.9** | 5.00 | 97.0 | **57.8** | 5.00 | 95.5 | **58.5** | 4.75 |
| Qwen2.5-14B(SFT) | Base | 76.0 | 45.4 | 5.63 | 74.9 | 35.1 | 6.35 | 76.3 | 40.6 | 6.39 | 74.0 | 38.0 | 6.15 |
| | Static+A | 94.1 | 60.6 | 4.69 | 88.7 | 41.3 | 5.24 | 87.9 | 41.3 | 5.28 | 86.9 | 42.4 | 5.08 |
| | Dynamic+A | 94.7 | 64.3 | 4.67 | 89.8 | 44.1 | 5.32 | 88.2 | 42.2 | 5.34 | 87.5 | 44.3 | 5.14 |
| | **Dynamic+H** | 93.2 | **71.0** | 4.61 | 90.2 | **48.3** | 5.17 | 87.6 | **44.5** | 5.14 | 87.3 | **47.8** | 4.98 |

Table 7: **Impact of NaviAgent Variants on API-Bank.** Metrics are reported as in Table 2.

## G LINK PREDICTION EVALUATION

| Dataset | APIs | Nodes | Edges | ACC | F1 | AUC |
|---|---|---|---|---|---|---|
| ToolBench | 5501 | 7866 | 24215 | 76.4 | 77.6 | 0.75 |
| API-Bank | 2650 | 6025 | 10255 | 78.4 | 76.1 | 0.71 |

Table 8: **Tool Graph Statistics and Link Prediction Evaluation.** Nodes and Edges denote the number of nodes and edges in the graph, respectively. ACC and F1 are reported as percentages (%), while AUC is reported as a value between 0 and 1.

## H USAGE OF LLM

To improve clarity and readability, we used a LLM for language polishing. All research ideas, methods, and conclusions were developed solely by the authors.

