# OpenReview forum: "NaviAgent: Bilevel Planning on Tool Navigation Graph for Large-Scale Orchestration"
_ICLR.cc/2026/Conference — Submitted to ICLR 2026_

### Official Review · Reviewer_JWrw · 2025-10-28

**Soundness:** 3
**Presentation:** 3
**Contribution:** 3
**Rating:** 6
**Confidence:** 4

**Summary:**

This paper presents NaviAgent, a bilevel framework for large-scale tool orchestration by LLMs. It decouples task planning (4D decision space: direct response, intent clarification, toolchain retrieval, execution) from tool execution (Tool World Navigation Model, TWNM). TWNM models tool structural/behavioral dependencies via a dynamic graph, enabling adaptive toolchain search. Closed-loop feedback optimizes planning/execution. Experiments on API-Bank/ToolBench show NaviAgent outperforms baselines in TSR, balancing efficiency/robustness .

**Strengths:**

1. This paper proposes a novel bilevel architecture that decouples task planning from tool execution, enabling NaviAgent to handle thousands of tools without being hindered by inter-tool complexity, thus addressing scalability issues of existing agents .
2. The Tool World Navigation Model (TWNM) dynamically encodes tool structural and behavioral dependencies, supports adaptive toolchain search/evolution, and significantly boosts performance on complex tasks by up to 17 points .
3. It integrates a closed-loop optimization mechanism using real tool interaction feedback to refine planning and execution, enhancing robustness and adaptability to dynamic API ecosystems .
4. For me, the strengths of this paper lie in the following aspects: it can dynamically adjust based on the difficulty level of various problems and the latest status when addressing them; it exhibits strong overall engineering feasibility; and in terms of innovation, the search strategies such as Alpha-Beta pruning can handle and prune some extreme cases, enabling rapid acquisition of effective toolchains.

**Weaknesses:**

1. My biggest concern is that this paper imposes overly strong constraints on input problems. For example, if we obtain the corresponding tool invocation path through the proposed graph, can we dynamically switch to an alternative path if a problem occurs in the middle of the current path? The paper seems to lack sufficient explanation regarding how to handle such errors.
2. When using Alpha-Beta pruning for search, the evaluation of Alpha and Beta values is crucial. For instance, if I choose a certain edge under a specific decision, how do you update the evaluation value of this decision in the global context? If only factors like tool invocation success rate and relevance are used, the accuracy of relevance evaluation needs to be very high. From this perspective, the paper’s evaluation of heuristic search seems relatively simplistic, and in some cases, it might eliminate effective tool invocation branches.
3. How do you evaluate the dependency between two tools? The paper mentions using weights for evaluation—are these weights based solely on the historical information of the tools observed so far? If the invocation relationship between two tools changes significantly, will the tools overly rely on historical data?
4. Does the proposed framework rely too heavily on the evaluation of tool invocation success rates? If a tool has a certain invocation success rate but delivers excellent results when it works, we might not should sacrifice its usage frequency. Alternatively, for your paper, is tool invocation speed more important compared to the overall reasoning performance?
5. Real-world requirements are highly diverse. For a new requirement, can this framework demonstrate better tool planning capabilities compared to traditional methods like ReACT and Tool-Planner?

**Questions:**

See above.

---

> ### Author Response · Authors · 2025-11-19
> **Response to Reviewer JWrw (Part 1)**
>
> We sincerely thank you for your thoughtful review and for recognizing our work’s key innovations in bilevel decoupling architecture, TWNM’s dynamic dependency modeling, and adaptive search mechanisms. Your insightful questions regarding real-world robustness, search evaluation, and error recovery provide valuable opportunities to further demonstrate the framework’s capabilities. Below we provide detailed responses to address your concerns comprehensively.
>
> **Q1: Dynamic path switching and error handling.**
>
> A1: Our framework supports dynamic replanning when an initially retrieved tool chain becomes infeasible or when any intermediate tool call fails during execution. In such cases, the NaviAgent automatically triggers the Re‑retrieval action to explore a new feasible subpath so that the task can continue without restarting.
>
> In the revised version of our paper, we have added a new figure after Section 3.3 to illustrate this process. The figure presents three representative modes of path switching: (1) Direct substitution: replacing a failed API with another that has identical functionality and compatible input–output schema; (2) Common‑output substitution: switching to one or multiple APIs whose combined outputs can fulfill the same intermediate requirements; (3) Goal‑level replanning: when the final target API becomes infeasible, the agent retrieves a new graph to complete the task through an alternative route. This figure clearly shows how TWNM performs replanning across these cases while maintaining task continuity and robustness without imposing stronger input constraints.
>
> **Q2: In the Alpha-Beta pruning algorithm, how does the framework update global decision evaluations beyond tool success and relevance to avoid pruning promising branches?**
>
> A2: In NaviAgent, edge evaluation is not solely based on instant success or relevance. Each time an edge is selected and executed, its outcome is stored in memory to guide subsequent planning. Its call statistics are updated immediately on the global tool graph, and the model that maps these statistics to edge weights is periodically retrained (Section 3.2.4). The updated weights feed into the composite fitness function of our heuristic search (Appendix B.1), which also considers node compactness, depth and complexity penalties, and path simplicity. Consequently, the global evaluation context is shaped by accumulated feedback rather than a single relevance estimate, while the hybrid heuristic search combining simulated annealing and genetic algorithms maintains stochastic exploration, reducing the risk of pruning promising branches.
>
> **Q3: How are dependencies between tools evaluated, and how does the model adapt if historical relationships change?**
>
> A3: TWNM defines the dependency between two tools through both structural and behavioral chains, as specified in Definition 1 and illustrated in the graph structure part of Figure 2. Structural chains capture parameter‑level connectivity between tools, while behavioral chains reflect their invocation relations observed in historical task executions. TWNM jointly learns from node semantics, structural relations, and invocation behaviors, ensuring that both newly added and updated tools acquire meaningful dependencies and remain well integrated into the overall network (see Eq. 13 in Appendix B.1). Consequently, the edge weights are not derived solely from historical statistics but are learned representations that encode both structural and behavioral information.
>
> This design also prevents over‑reliance on outdated patterns when invocation relationships change. Structural modifications automatically trigger the incremental node‑integration mechanism, which updates input/output schemas and resets the usage statistics of newly added nodes to zero. In addition, the graph‑evolution process periodically updates edge weights to balance long‑term stability and short‑term success feedback. The tunable coefficient η (see Eq. 12 in Section 3.2.4) controls the model’s sensitivity to recent dynamics, thereby ensuring timely adaptation while preserving overall consistency.

---

> ### Author Response · Authors · 2025-11-19
> **Response to Reviewer JWrw (Part 2)**
>
> **Q4: Dependence on success‑rate evaluation, treatment of low‑success but high‑impact tools, and whether the framework prioritizes speed over reasoning performance.**
>
> A4: As discussed in Response Q3, TWNM learns each edge weight  jointly from semantic, structural, and behavioural information rather than relying on any single metric. Among these factors, the invocation success rate acts as one reliability cue that complements other statistics such as structural degrees and historical invocation counts (see Eq. 13 and Figure 1). It contributes to but does not dominate the learning process. During graph search and pruning (Figure 2, Eq. 18–19), TWNM evaluates candidate paths using a composite score that integrates structural compactness, path depth, cumulative edge weights, and complexity constraints. This design ensures that tools with lower success rates are not automatically eliminated. Such tools can still receive comparatively high learned weights if they are frequently invoked or occupy structurally important positions in the tool graph, allowing them to remain active during candidate subgraph exploration. In summary, our framework primarily aims to improve reasoning robustness and compositional success, while runtime efficiency naturally improves as the model learns more concise reasoning paths.
>
> **Q5: Does your framework achieve better tool planning on new requirements than ReACT and Tool‑Planner?**
>
> A5: NaviAgent is capable of handling diverse real‑world requirements. When a new task or domain introduces unseen tools or parameters without historical usage data, the TWNM dynamically incorporates them into the existing graph through the Incremental Node Integration mechanism (Section 3.2.4). The merging decision for a new tool is based on its functional description and input–output schema, while the new node’s representation is initialized as a weighted average of its connected neighbors, enabling the graph to expand adaptively while preserving structural consistency and avoiding immediate retraining. In cold‑start experiments, where all representations are initialized solely from functional descriptions, the Static + A configuration outperformed the no‑graph baseline by about 6.5 percentage points  (Table 2), demonstrating that NaviAgent effectively reasons over unseen tools using structural and semantic information.
>
> Compared with ReACT, which invokes tools sequentially without modeling their relationships, NaviAgent performs reasoning and planning over a structured dependency graph that explicitly captures how different tools cooperate along a task chain. This leads to automatic multi‑tool plan construction and efficient error recovery, achieving on average a 14.5‑point higher task success rate (Table 1 vs. Table 2). Tool‑Planner, as cited in our Introduction (Liu et al., 2024c), provides a valuable substitution mechanism within homogeneous toolkits. NaviAgent builds on this idea and extends it to large‑scale, heterogeneous tool ecosystems where tools differ in both function and dependency. By jointly considering functional similarity, input–output compatibility, and cross‑functional dependencies (e.g., connecting a web‑search tool to a summarization module), NaviAgent demonstrates stronger planning capability and better generalization to new requirements.

---

> > ### Comment · Reviewer_JWrw · 2025-11-20
> > **Thanks for your response**
> >
> > Thanks for your insightful response, I would update my evaluation to 8.

---

> > > ### Author Response · Authors · 2025-11-20
> > > **Thanks for your positive feedback**
> > >
> > > We sincerely thank you for your time, thoughtful feedback, and the improved evaluation. Your constructive comments helped us further refine the details and clarity of our paper, and we truly appreciate your recognition of our work.

---

### Official Review · Reviewer_XNiU · 2025-10-31

**Soundness:** 2
**Presentation:** 3
**Contribution:** 3
**Rating:** 6
**Confidence:** 2

**Summary:**

The paper presents NaviAgent, a bilevel planning framework for tool-use agents. It separates high-level reasoning (deciding when to respond, clarify, retrieve, or execute) from low-level execution using a Tool World Navigation Model (TWNM), a dynamic graph that captures dependencies among tools. Their quantitative experiments show significant gains in task success and completion rates over baselines.

**Strengths:**

The research problem is timely and important, given the rise of agentic LLMs

The quantitative experiments show consistent improvements over the baselines.

**Weaknesses:**

1. The paper provides no qualitative or quantitative analysis of the learned graph structure; TWNM is evaluated only indirectly through overall task performance, making it difficult to see what the graph actually learns.

2. The high-level decision labels (Direct Response / Clarify / Retrieve / Execute) are derived from rule-based relabeling of ToolBench and API-Bank traces with additional synthetic augmentation, rather than real human data. This raises concerns about whether the learned planner generalizes to real-world cases, particularly in the Direct Response and Clarify categories.

**Questions:**

1. In Section 3.3 (L319–323), the paper mentions repeating recombination “until infeasibility,” but the stopping criterion is not defined. Could the authors clarify how termination is determined?

2. How sensitive is the planner to the 4 action taxonomy? Could a different or better set of decision type help?

3. The paper claims TWNM supports dynamic integration of new tools; do you have evidence for tool generalization unseen during training?

4. How expensive is maintaining and updating TWNM as tool set grows?

---

> ### Author Response · Authors · 2025-11-19
> **Response to Reviewer XNiU (Part 1)**
>
> We sincerely appreciate your time and your acknowledgment of this important research problem in complex tool invocation scenarios.  Your insightful questions concerning TWNM’s graph structure analysis, the action taxonomy’s validity, and the framework’s scalability provide valuable perspectives, as they highlight key considerations for both theoretical rigor and practical deployment.  We address each of these points in detail below.
>
> **Q1: The paper lacks qualitative or quantitative analysis of the learned graph structure.**
>
> A1: In our paper, we present both qualitative and quantitative analyses that directly reveal what TWNM learns from tool interactions and how the learned graph evolves over time. As shown on the right side of Figure 2, we visualize its evolution at three timestamps (T−1, T, T+1), where edges with higher weights are drawn in darker colors. For clearer presentation of tool dependencies, parameter nodes are omitted. Initially, selectable paths are almost uniform, indicating multiple equivalent routes. Over time, TWNM learns to emphasize more successful tool combinations (the darker top path at T) while down‑weighting unreliable ones. When the upper‑left API fails at T+1, the system recomputes edge weights and automatically redirects to a viable subpath, indicating adaptation to feedback.
>
> These qualitative observations are supported by quantitative results. We evaluate the learned graph structure on ToolBench and ApiBank (Table 8, Appendix G). The graph embeddings achieve 76–78 percent accuracy and about 77 percent F1 (AUC > 0.7), showing that the learned representations effectively predict plausible tool relations rather than random links. Together, these results show that TWNM learns meaningful and adaptive dependencies among tools.
>
> **Q2: Concern about generalization of rule-based four decision labels, especially Direct Response and Clarify.**
>
> A2: The four decision categories are not rule‑generated labels, but a deliberately designed decision space derived from extensive experience in large‑scale tool‑use scenarios. They capture the essential reasoning phases observed in practice: respond, clarify, retrieve, and execute.
>
> Among them, the Direct Response and Clarify decisions are naturally present in real user interactions, not artificially induced by labeling rules. Direct Response occurs when the agent already has enough information from its own knowledge or the current context to answer the query without using any tools. Clarify is triggered when the user input is ambiguous or incomplete and additional information is needed before selecting a tool. Both behaviors are crucial for maintaining interaction efficiency and accuracy. Across diverse domains (e.g., weather, healthcare, finance), our framework shows consistent improvements on ToolBench (Table 1), APIBank (Table 7), and real-world API evaluations (Table 5), demonstrating robust and consistent performance across tasks.
>
> **Q3: Clarify termination condition in recombination process.**
>
> A3: Thank you for pointing this out. We have clarified in Section 3.3 that the recombination process terminates after at most four iterations (i.e., four recombination steps), or earlier if no feasible combination can be found. This upper limit provides a clear stopping condition and avoids unnecessary computation.
>
> **Q4: How sensitive is the planner to the 4 action taxonomy? Could a different or better set of decision type help?**
>
> A4: The four action types form a minimal yet sufficient set covering two complementary aspects of tool use: user interaction and tool operation. As shown in Figure 6 (see Section 4.3), Clarification and Re‑retrieval (an adaptive form of ToolChain Retrieval) improve success rates across difficulty levels compared with baseline frameworks such as ReAct, ToolLLM, and α‑UMI, confirming the stability and practical effectiveness of this taxonomy. This work primarily focuses on developing a framework for large‑scale, compositional tool use that achieves reliable performance gains.

---

> ### Author Response · Authors · 2025-11-19
> **Response to Reviewer XNiU (Part 2)**
>
> **Q5: Do you have evidence for tool generalization unseen during training?**
>
> A5: In our setting, unseen tools refer to those without any historical usage data but with known functional descriptions and input–output schema. TWNM supports the integration of such tools through the Incremental Node Integration mechanism (Section 3.2.4). When a new tool is introduced, it is initialized as a weighted average of its neighbors’ representations, ensuring structural consistency after expansion and avoiding immediate retraining.
>
> To directly evaluate generalization in this scenario, we conduct an experiment where all tool representations are initialized solely from their functional descriptions, without relying on any prior usage data. As shown in Table 2 (Section 4.3), the Static + A variant outperforms the no‑graph baseline by about 6.5 percentage points on average, indicating that TWNM effectively integrates and reasons over previously unseen tools.
>
> **Q6: How expensive is maintaining and updating TWNM as tool set grows?**
>
> A6: TWNM does not immediately retrain the TWNM upon each incremental node update. Instead, the graph weights are periodically updated to maintain long‑term stability while still incorporating short‑term feedback.  A tunable coefficient η balances recent and historical signals (Eq.12, Section 3.2.4), allowing controllable responsiveness when higher adaptivity is needed. The update interval is flexibly scheduled, typically once every one week according to service load, which keeps the maintenance cost predictable and stable as the tool set expands.
>
> At the current scale of about 5.5K tools, 7.9K nodes, and 24K edges (Table 8 in Appendix G), an offline full‑graph retraining takes about 94s on a single NVIDIA P40 GPU, after which the new graph is hot‑swapped into the online database with negligible serving latency. Incremental operations are much lighter: hot‑node patches (updates of tool usage statistics) take around 55 ms, and adding new tools, including semantic matching and attribute initialization, completes within 100 ms per insertion. The full‑graph update runs offline and does not affect online tool calls, while incremental updates execute in real time without noticeable service slowdown. In deployment, the graph occupies 1.5 GB memory on a 4‑core CPU / 8GB RAM server. The current graph already covers a large‑scale production environment with thousands of tools, yet maintains low computational and operational overhead. These results suggest that the maintenance cost of TWNM remains practical and stable even as the tool set continues to expand.

---

> > ### Comment · Reviewer_XNiU · 2025-11-27
> >
> > Thank you for the responses. I will keep my score.

---

> > > ### Author Response · Authors · 2025-11-27
> > >
> > > Thank you for your review and for the time you spent providing valuable feedback. It is very helpful for improving our work.

---

### Official Review · Reviewer_JzGE · 2025-11-01

**Soundness:** 3
**Presentation:** 3
**Contribution:** 2
**Rating:** 4
**Confidence:** 3

**Summary:**

The paper proposes the NaviAgent bilevel framework, which decouples task planning and execution in LLM tool orchestration. Integrating TWNM (Tool World Navigation Model) to dynamically model tool dependencies and enable closed-loop optimization, NaviAgent outperforms baselines significantly in Task Success Rate (TSR) on the API-Bank and ToolBench datasets, achieving efficient navigation in large-scale tool ecosystems.

**Strengths:**

1.  Its architectural innovation decouples key components (planning and execution).
2.  The  TWNM design unifies the capture of both tool structural and behavioral dependencies.
3.  The paper have conducted comprehensive experiments cover multiple models and scenarios.
4. The paper is well-written, featuring a clear logical structure.

**Weaknesses:**

1.  There is a gap between tools in simulated and real environments. Real-world APIs are diverse and dynamic, with frequent error fixes, feature updates (e.g., new parameters added), or temporary outages. Although TWNM incorporates a dynamic graph evolution design, it relies on historical execution feedback to update the graph structure, leading to a time gap—for instance, if an API’s error is just fixed but its weight in the graph remains low, NaviAgent may still avoid using it; conversely, sudden API failures without timely pruning result in invalid calls.
2.  Real-world APIs are extensive, and numerous tools not included in the initial graph exist, causing TWNM to fail in generating optimal toolchains.

**Questions:**

please refer to Weaknesses

---

> ### Author Response · Authors · 2025-11-13
> **Response to Reviewer JzGE**
>
> We sincerely appreciate your positive assessment of our architectural innovations in decoupling planning and execution, as well as TWNM’s comprehensive approach to modeling tool dependencies. Regarding the challenges of evolving real-world APIs, which represent a common challenge for tool-use systems, our framework explicitly accounts for this open‑world and dynamic nature of tools.
>
> **Q1: How does TWNM handle rapid API changes and updates in real environments?**
>
> A1: Our framework addresses these API dynamics through two complementary mechanisms. First, after each dialogue episode, TWNM immediately updates the node attributes, such as a tool’s recent success rate and usage frequency, based on the latest execution results.  These statistics are then used in the next pruning step so that tools with lower recent performance are down‑weighted or pruned promptly. Second, the graph weights are periodically updated to reflect longer‑term stability while still incorporating short‑term success feedback. The balance between recent and historical observations is controlled by a tunable coefficient η (see Eq. 12, Section 3.2.4), allowing the system to adjust its sensitivity to short‑term dynamics when higher responsiveness is desired.
>
> To illustrate this idea, consider a weather API that can be called using different input forms such as city name, geographic coordinates, or zip code to retrieve the same type of information. NaviAgent dynamically adjusts the weight of each form based on recent execution statistics, favoring more reliable inputs. It also performs adaptive path re‑routing when tool calls fail, and an additional Figure 3  (Section 3.3)  illustrates this replanning process. Real‑API experiments (Appendix F.1) further demonstrate higher execution accuracy, and fewer invocation steps than SOTA baselines such as a‑UMI.
>
>
> **Q2: How can TWNM adapt to unseen or newly added tools beyond the initial graph?**
>
> A2: TWNM dynamically incorporates new or modified tools through incremental node integration, allowing the graph to evolve alongside the real‑world tool ecosystem. When a new API emerges or an existing one changes, the framework connects or adjusts its corresponding node in the graph based on the semantic and structural similarity of its input–output schemas. If a closely related node is found, its link is updated; otherwise, a new node is initialized. Its attributes (e.g., success/failure counters and statistical edge weights) are set to zero, and its embedding is derived from a weighted average of neighboring nodes to maintain local structural consistency  (Section 3.2.4).
>
> During subsequent graph updates, TWNM jointly learns from node semantics, structural relations, and invocation behaviors, ensuring that both newly added and existing nodes form meaningful dependencies within the overall network  (Eq. 13, Appendix B.1). This enables the tool graph to evolve continuously with real‑world API dynamics, avoiding any “closed‑world” limitation. In the ablation experiments (Table 2), the dynamic‑graph variants (Dynamic +A / Dynamic + H) achieve the best overall performance, while the static variant (Static +A), initialized only from tool functionalities and I/O schemas without historical data, still outperforms the base configuration without TWNM by an average of  6.5 percentage points. These results confirm that TWNM generalizes effectively to unseen tools and maintains near‑optimal toolchain generation as the ecosystem evolves.

---

### Official Review · Reviewer_4eEC · 2025-11-01

**Soundness:** 3
**Presentation:** 3
**Contribution:** 3
**Rating:** 6
**Confidence:** 4

**Summary:**

The paper addresses the challenges LLM-based agents face when orchestrating large-scale, dynamic tool ecosystems, specifically targeting issues arising from sequential, step-by-step tool invocation, such as error accumulation and inability to handle complex inter-tool dependencies. The authors propose NaviAgent, a bilevel framework that decouples high-level task planning from low-level tool execution.

**Strengths:**

- **Bilevel Decoupling:** The core architecture uniquely decouples high-level task reasoning (the four-dimensional decision space: respond, clarify, retrieve, execute) from low-level tool orchestration. This contrasts with standard ReAct-style agents that interleave reasoning and single-step execution, often leading to error accumulation in complex tasks.
- **Evolving Tool World Navigation Model (TWNM):** Moving beyond static tool graphs, the TWNM is highly original in its integration of "behavioral chains" derived from actual execution traces alongside standard "structural chains" (API schemas). Treating inter-tool dependency discovery as a link prediction problem using Heterogeneous Graph Transformers is a sophisticated approach to a typically heuristic-heavy domain.
- **Search Algorithms for Toolchains:** The adaptation of classic search algorithms—specifically Alpha-Beta pruning for backward search and a hybrid genetic/simulated annealing heuristic for forward search—to orchestrate entire toolchains is a creative departure from standard retrieval-augmented generation (RAG) or simple depth-first search methods.

**Weaknesses:**

- **Cold-Start**: The Tool World Navigation Model (TWNM) heavily relies on "behavioral chains, derived from historical usage data" and "statistical weight... reflecting empirical invocation patterns". This creates a significant cold-start problem. The framework might underperform significantly in new domains where these rich execution traces are unavailable, yet the paper does not quantify this degradation.
- **Justifications of Four-Dimensional Decision Space**: The high-level planner uses a fixed "four-dimensional decision space" (Direct Response, Intent Clarification, ToolChain Retrieval, Tool Execution). While functional, it is not clear if this specific taxonomy is optimal or necessary compared to a more flexible, LLM-driven dynamic planning approach. It risks being too rigid for edge cases that don't fit neatly into these four categories (e.g., partial execution with mid-stream replanning without full re-retrieval).
- **No Optimization for Arguments**: For tool function calling tasks, selecting the correct function is important. But most time, LLMs usually fail at using the correct arguments for the functions.

**Questions:**

See Weaknesses.

---

> ### Author Response · Authors · 2025-11-19
> **Response to Reviewer 4eEC**
>
> Thank you for your thorough and constructive review. We sincerely appreciate your recognition of our work’s novel contributions, including the bilevel decoupling architecture, the Tool World Navigation Model, and the adaptive search algorithms. We also value your insightful questions regarding cold-start scenarios, the four-dimensional decision space, and argument optimization. We have carefully addressed each of these points in the following responses.
>
> **Q1: Can TWNM effectively address cold-start issues in new domains?**
>
> A1: Cold‑start robustness is an important consideration in the design of TWNM. To systematically evaluate the framework under settings without historical data, we include a Static + A experiment in our paper (see Table 2). In this setting, all historical behavioral chains and dynamic statistical weights are removed, and the model can only rely on the static structural connections between tools, including input–output parameter relationships and functional descriptions represented as node embeddings.
>
> This configuration precisely simulates the initial deployment phase in a new domain where no execution history is available. The results show that the static graph’s performance is only 2.6 percentage points lower than the dynamic graph variant (Dynamic + A), which uses full historical data. This quantifies the effect of the cold‑start condition. Moreover, even under these extreme conditions, the static graph still outperforms the no‑graph baseline by 6.5 percentage points on average with fewer reasoning steps. These findings demonstrate that the design of TWNM provides strong navigation and reasoning capabilities even without historical data.
>
> **Q2: Whether fixed four-category planner is too rigid or suboptimal for flexible planning.**
>
> A2: The high‑level planner balances interpretability and flexibility through four distinct planning actions. ReAct is a representative form of flexible LLM‑driven dynamic planning. It interleaves reasoning and acting across multiple steps but lacks an explicit planning taxonomy. Building on this insight, our framework organizes these decision patterns into four actions: Direct Response, Intent Clarification, ToolChain Retrieval, and Tool Execution. This design preserves the flexibility of ReAct while providing a clearer planning structure that enhances transparency in complex tasks.
>
> As shown in Figure 6, the framework achieves a 9.3 percent higher task success rate than ReAct on average. Further analysis shows that the improvement mainly comes from the additional Clarification and Re‑retrieval phases, confirming that the four-dimensional decision space provides a complete and effective framework for tool reasoning. In particular, the planner demonstrates flexible re‑retrieval behavior when a tool execution step fails. In this case, it re‑invokes the ToolChain Retrieval action to construct an alternative sub‑chain near the failure point. To illustrate this process more clearly, we have added Figure 3 in Section 3.3, which depicts three representative modes of path switching during replanning: direct substitution, common‑output substitution, and goal‑level replanning. This figure highlights how the planner maintains continuity and robustness without enforcing rigid input constraints.
>
> **Q3: LLM often fails to produce correct function arguments in tool use.**
>
> A3: Argument quality is explicitly handled in our tool execution stage. We apply a lightweight post‑processing step that (i) maps LLM‑generated argument names to the closest valid ones defined in the tool schema based on string‑overlap matching, and (ii) converts argument values to their expected data types. This ensures reliable function invocation even when the llm’s raw arguments are imperfect.

---

> > ### Comment · Reviewer_4eEC · 2025-11-20
> >
> > Thank you for your reply. My previous question has been solved and I improve my score accordingly.

---

> > > ### Author Response · Authors · 2025-11-21
> > >
> > > We appreciate your time and the positive update. We are pleased that our responses resolved your concerns and thank you for acknowledging our work.

---

### Author Response · Authors · 2025-12-02
**General Response**

Dear ICLR 2026 AC, SAC, and PC,

We sincerely thank all reviewers for their valuable time, constructive feedback, and consistent recognition of our work’s novelty, a bilevel framework that decouples high‑level planning from low‑level execution. This design introduces global task planning to reduce cascading errors and enable scalable multi‑tool reasoning, especially under large‑scale tool orchestration scenarios.

The reviews mainly focus on how the framework behaves in evolving and unseen tool environments, which is a common challenge in the tool‑use domain and also the main motivation for our work. Our framework addresses these concerns through three core contributions:

1. **Flexible planning**. The high‑level planner defines a four‑dimensional action space, which forms a minimal yet sufficient set covering user interaction and tool operation. As shown in Figure 6 and Tables 1-2, Clarification and Re‑retrieval yield a 14.5 point higher task success rate than ReAct and outperform other baselines (ToolLLM, $\alpha$‑UMI).

2. **Dynamic tool adaptation**. TWNM captures both structural and behavioral dependencies among tools and employs a graph evolution mechanism that balances short‑term feedback with long‑term stability. In table 2, the dynamic variant performs best, and the static graph (without any historical data) still outperforms the no‑graph baseline by 6.5 points with only a 2.6‑point gap to the dynamic setting, demonstrating adaptability to new tools and domains in an evolving tool ecosystem.

3. **Robust error recovery**. When a retrieved tool chain becomes infeasible or an intermediate tool call fails, NaviAgent automatically triggers the Re‑retrieval action to generate a new feasible subpath so that the task can continue without restarting. We have added Figure 3 in the revised paper to illustrate this process in detail.

As noted by Reviewer JzGE,  our initial experimental settings are comprehensive, covering large-scale tool benchmarks (API-Bank and ToolBench) and real API testing. Designed for production-level tool scenarios, our approach also includes an explicit computational cost analysis (see A6 response to Reviewer XNiU) for real‑world reference.

During the discussion phase, **Reviewers JWrw and 4eEC both raised their scores (from 6 to 8) before Nov 22**, further acknowledging our work. Reviewer JzGE did not participate. We kindly hope you will consider both the contributions and the practical applicability of this paper to the tool‑learning community. Thank you for your time and effort.

Best regards,

The authors of Paper 16319

---

### Meta-Review · Area_Chair_bpUi · 2026-01-06

**Summary:**

This paper presents NaviAgent, a system designed to help AI agents use thousands of different tools by separating high-level planning from the actual execution of the tools. It uses a graph-based model to understand how different tools relate to one another. While the reviewers liked the idea of a "bilevel" architecture and the use of search algorithms like Alpha-Beta pruning, the paper is being suggested for rejection primarily due to concerns about how it would work in the real world. Specifically, there is a large gap between the controlled, simulated environments used in the experiments and the messy, constantly changing nature of real-world APIs.

**Reviewer Concerns:**

During the discussion, the authors provided more details on how the system handles new tools and shared a cost analysis. This led two reviewers to raise their scores significantly. However, several critical concerns remain outstanding. Reviewer JzGE pointed out that real-world APIs change much faster than the graph can update, leading to a "time gap" where the agent might try to use broken tools. Additionally, Reviewer XNiU highlighted that the high-level planning categories were created using simple rules and synthetic data, rather than real human interactions. This makes it unclear if the agent would actually know how to "clarify" or "respond" correctly when talking to a real person in a complex situation.

**Reviewer Scores:**

Reviewers JWrw and 4eEC both moved their scores to an 8 (Accept), feeling that the rebuttal addressed their technical questions about path-finding and cold-start issues. However, Reviewer JzGE, who provided the most critical score of 4, did not participate in the final discussion. If they had participated, it is unlikely they would have changed their mind, as the authors' response about "periodic updates" does not fully solve the problem of sudden API failures. Reviewer XNiU stayed at a 6, signaling that while the work is interesting, the lack of real-world data is a major limitation.

---

### Decision · Program_Chairs · 2026-01-26

Reject